# Vad-R1: Towards Video Anomaly Reasoning via Perception-to-Cognition Chain-of-Thought

**Chao Huang**[1]    **Benfeng Wang**[1]    **Wei Wang**[1][*]    **Jie Wen**[2]    **Chengliang Liu**[3]
**Li Shen**[1,4]    **Xiaochun Cao**[1]

[1]Shenzhen Campus of Sun Yat-sen University [2]Harbin Institute of Technology, Shenzhen
[3]Laboratory for Artificial Intelligence in Design, The Hong Kong Polytechnic University
[4]Shenzhen Loop Area Institute

{huangch253, wangwei29, caoxiaochun}@mail.sysu.edu.cn   wangbf23@mail2.sysu.edu.cn
wenjie@hit.edu.cn   liucl1996@163.com   mathshenli@gmail.com

## Abstract

Recent advancements in reasoning capability of Multimodal Large Language Models (MLLMs) demonstrate its effectiveness in tackling complex visual tasks. However, existing MLLM-based Video Anomaly Detection (VAD) methods remain limited to shallow anomaly descriptions without deep reasoning. In this paper, we propose a new task named Video Anomaly Reasoning (VAR), which aims to enable deep analysis and understanding of anomalies in the video by requiring MLLMs to think explicitly before answering. To this end, we propose Vad-R1, an end-to-end MLLM-based framework for VAR. Specifically, we design a Perception-to-Cognition Chain-of-Thought (P2C-CoT) that simulates the human process of recognizing anomalies, guiding the MLLMs to reason about anomalies step-by-step. Based on the structured P2C-CoT, we construct Vad-Reasoning, a dedicated dataset for VAR. Furthermore, we propose an improved reinforcement learning algorithm AVA-GRPO, which explicitly incentivizes the anomaly reasoning capability of MLLMs through a self-verification mechanism with limited annotations. Experimental results demonstrate that Vad-R1 achieves superior performance, outperforming both open-source and proprietary models on VAD and VAR tasks. Codes and datasets will be released at https://github.com/wbfwonderful/Vad-R1.

## 1 Introduction

Video Anomaly Detection (VAD) focuses on identifying abnormal events in videos, and has been widely applied in a range of domains like surveillance systems [53] and autonomous driving [39, 79]. Traditional VAD methods typically fall into two paradigms: semi-supervised and weakly-supervised VADs. The semi-supervised VAD methods [79, 34, 20, 36, 19, 17, 22] aim at modeling the features of normal events, while there are only video-level annotations available for weakly-supervised VAD methods [71, 53, 18, 17, 86, 26, 21, 70]. With the development of vision-language models, some studies introduce semantic information into VAD [64, 73, 72, 80, 7, 23]. However, traditional VAD methods only remain at the level of detection, lacking understanding and explanation of anomalies.

Recently, the reasoning capability of large language models has emerged as a key frontier [44, 9, 58]. Unlike daily dialogue, reasoning requires models to think before answering, enabling them to perform causal analysis and further understanding. In particular, DeepSeek-R1 demonstrates the effectiveness of Reinforcement Learning (RL) in stimulating and refining reasoning capability [9]. Concurrently, parallel efforts have begun to extend reasoning to the multimodal domain [57, 60].

---

[*]Corresponding author.

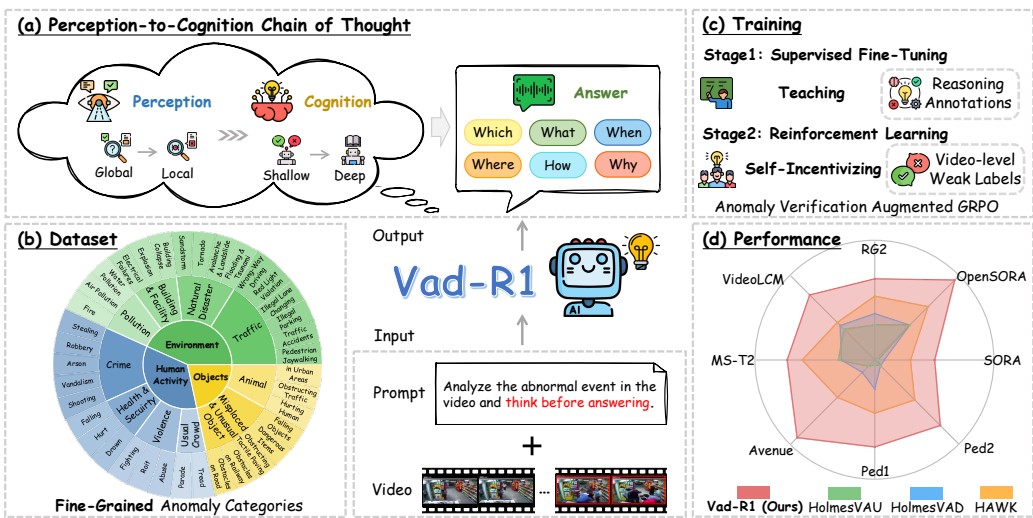

Figure 1: **Overview of Vad-R1.** Vad-R1 is an end-to-end framework for video anomaly reasoning. A structured Perception-to-Cognition Chain-of-Thought is introduced to guide Vad-R1 in performing step-by-step reasoning. Based on the structured CoT, a new dataset for video anomaly reasoning is constructed, including fine-grained anomaly categories. Then, A two-stage training pipeline is adopted to progressively enhance the reasoning capability of Vad-R1. Finally, Vad-R1 outperforms existing MLLMs-based VAD methods with a great margin on VANE benchmark.

Despite the growing interest in reasoning capability, existing Multimodal Large Language Models (MLLMs) based VAD methods still fall short in this regard. Those methods can be divided into two categories based on the role of MLLMs. Some methods regard MLLMs as auxiliary modules [38, 88, 89, 11], where MLLMs provide supplementary explanation after the classifier predicts the anomaly confidence. In this context, anomaly understanding is a step after detection, and the output of MLLMs does not directly promote anomaly detection. Subsequently, although some methods utilize MLLMs to directly perform anomaly detection and understanding [54, 40, 77, 83, 13, 12], MLLMs only generate anomaly descriptions or perform simple anomaly question answering based on video content, lacking thinking and analytical abilities. Thus, reasoning remains underexplored in VAD.

To bridge this gap, we propose a new task: Video Anomaly Reasoning (VAR), which aims to empower MLLMs with the ability to perform structured, step-by-step reasoning about anomalous events in the videos. Compared with existing video anomaly detection or understanding tasks, VAR targets a deeper level of analysis by mimicking the human cognitive process, enabling contextual understanding, behavior interpretation, and norm violation analysis. To this end, we propose Vad-R1, the first end-to-end MLLM-based framework for VAR, which explicitly performs reasoning before generating its response. However, enabling reasoning in video anomaly tasks presents two major challenges. Firstly, existing VAD datasets lack structured reasoning annotations, making them unsuitable for training and evaluating anomaly reasoning models. Secondly, how to effectively train models to acquire reasoning capability remains an open challenge. Unlike tasks with clearly defined objectives, open-ended VAR requires models to perform multi-step causal reasoning, making it difficult to define clear training objectives and directly guide the reasoning process.

**For the first challenge**, we design a structured Perception-to-Cognition Chain-of-Thought (P2C-CoT) for video anomaly reasoning, as shown in Figure 1(a). Inspired by the process of human understanding the anomalies in the videos, the proposed P2C-CoT first guides the model to perceive from the global environment of the video to the suspicious clips of the video. After perception, the model will make cognition based on visual clues from shallow to deep level. Finally, the model gives the analysis result as answer, including the anomaly category, the anomaly description, the temporal range of anomaly, the approximate spatial position of the anomaly and so on. Then based on the CoT, we construct Vad-Reasoning, a specially designed dataset for VAR, which includes fine-grained anomaly categories as shown in Figure 1(b). Vad-Reasoning consists of two complementary subsets. One subset contains videos with P2C-CoT annotations, which are generated by proprietary models step-by-step. The other subset contains a larger number of videos, where there are only video-level

weak labels available due to high annotation costs. **For the second challenge**, inspired by the success of DeepSeek-R1, we propose a training pipeline with two stages as shown in Figure 1(c). In the first stage, Supervised Fine-Tuning (SFT) is performed to equip the base MLLM with fundamental anomaly reasoning capability. In the second stage, RL is employed to further incentivize the reasoning capability with the proposed Anomaly Verification Augmented Group Relative Policy Optimization (AVA-GRPO) algorithm, an extension of original GRPO [50] specifically designed for VAR. During RL training, the model first generates a group of completions. Based on these completions, the original videos are temporally trimmed and the trimmed videos are then fed back to the model to generate new completions. The two sets of completions are subsequently compared, and an additional anomaly verification reward is assigned if a predefined condition is satisfied. Finally, AVA-GRPO promotes MLLM's video anomaly reasoning capability through this self-verification mechanism with limited annotations. In summary, the contributions of this paper are threefold:

- We propose a new task named Video Anomaly Reasoning (VAR), which extends traditional VAD from surface-level recognition to deeper cognitive understanding. To this end, we develop Vad-R1, a novel end-to-end MLLM-based framework tailored for VAR, which aims at further analysis and understanding of anomalies in the video.

- We design a structured Perception-to-Cognition Chain-of-Thought to guide the model in performing step-by-step structured reasoning. Building upon this paradigm, we construct Vad-Reasoning, a specially designed dataset for video anomaly reasoning with two subsets. Furthermore, we propose an improved reinforcement learning algorithm AVA-GRPO, which incentivizes the reasoning capability of MLLMs through a self-verification way.

- The experimental results demonstrate that the proposed Vad-R1 achieves consistently superior performance across multiple evaluation scenarios, surpassing both open-source and proprietary models in video anomaly detection and reasoning tasks.

## 2 Related Works

**Video Anomaly Detection and Dataset**    Video anomaly detection aims at localizing the abnormal events in the videos. Based on the training data, traditional VAD methods typically fall into two paradigms, the semi-supervised VAD [79, 34, 20, 36, 19, 17, 48] and weakly supervised VAD [71, 53, 18, 17, 26, 21]. Furthermore, some studies try to introduce text description to enhance detection [64, 73, 72, 80, 7, 8]. Recently, there has been growing interest in integrating MLLMs into VAD to improve understanding and explanation [38, 54, 40, 77, 83, 88, 89, 11, 13, 12]. However, current studies remain at shallow understanding with MLLMs, lacking in-depth exploration of reasoning capability. In this paper, we propose an end-to-end framework to explore the enhancement of reasoning capability for video anomaly tasks.

Furthermore, the existing VAD datasets primarily provide coarse-grained category labels [53, 71, 39, 1] or abnormal event description [13, 12, 54, 82], lacking annotation of reasoning process. To address this gap, we propose a structured Perception-to-Cognition Chain-of-Thought and a dataset specially designed for video anomaly reasoning, providing step-by-step CoT annotations.

**Video Multimodal Large Language Model**    The video multimodal large models provide an interactive way to understand video content. Early works integrate visual encoders into large language models by aligning visual and textual tokens via mapping networks or adapter layers [27, 32, 41, 87, 91]. Compared with static images, videos inherently contain rich temporal dynamics and redundant visual information. To address this, some studies explore token compression mechanism to handle longer contexts and reduce computational overhead [31, 76, 90, 25]. In addition, recent works have explored online video stream understanding [6, 10, 78, 74]. Nevertheless, these methods remain at the level of video understanding and lack exploration of reasoning capability.

**Multimodal Large Language Model with Reasoning Capability**    Enhancing the reasoning capability of MLLMs has become a major research focus. Early studies have explored multi-stage reasoning frameworks and large-scale CoT datasets to enhance the reasoning capability of MLLMs [75, 63, 35]. Recently, DeepSeek-R1 [9] demonstrates the potential of reinforcement learning in enhancing the reasoning capability, inspiring subsequent efforts to reproduce its success in multimodal domains [24, 84]. In the field of video understanding, several studies also utilize RL

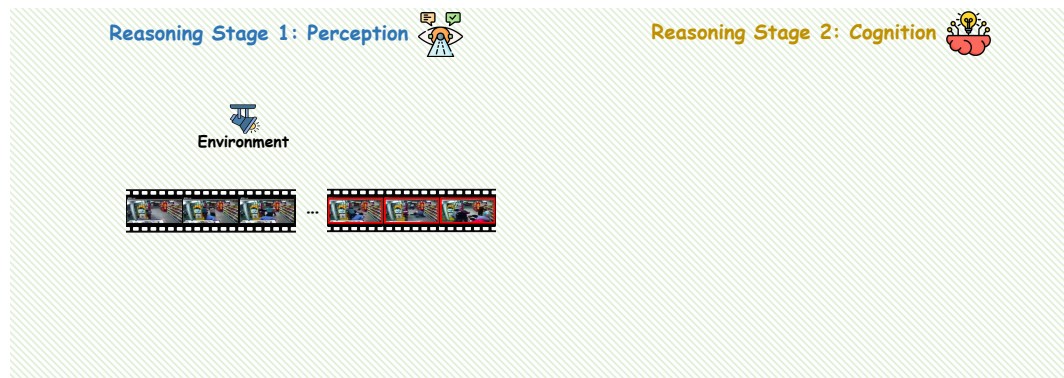

(a) Illustration of the proposed structured Chain-of-Thought, including two stages: perception and cognition.

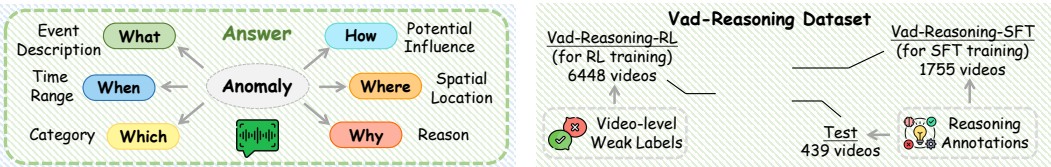

(b) Illustration of the answer after reasoning.

(c) The arrangement of Vad-Reasoning dataset.

Figure 2: Overview of the proposed Perception-to-Cognition CoT and Vad-Reasoning dataset.

to improve spatial reasoning [30], temporal reasoning [68] and general causal reasoning [14, 92]. Building upon these advances, this paper focuses on the video anomaly reasoning task.

## 3 Method: Vad-R1

**Overview** In this section, we introduce Vad-R1, a novel end-to-end MLLM-based framework for VAR. The reasoning capability of Vad-R1 is derived from a two-stage training strategy: SFT with high quality CoT annotated videos and RL based on AVA-GRPO algorithm. We begin by introducing the proposed P2C-CoT in Section 3.1. Based on the P2C-CoT, we construct Vad-Reasoning, a new dataset as detailed in Section 3.2. Then, we introduce the improved RL algorithm AVA-GRPO in Section 3.3. Finally, we introduce the training pipeline of Vad-R1 in Section 3.4.

### 3.1 Perception-to-Cognition Chain-of-Thought

When humans interpret a video, they typically first observe the events that occur in the video, and then develop a deeper understanding based on visual observation. Motivated by this, we design a structured Perception-to-Cognition Chain-of-Thought (P2C-CoT) for video anomaly reasoning, which gradually transitions from **Perception** to **Cognition** consisting of 2 stages with 4 steps as shown in Figure 2(a), and concludes with a concise answer as shown in Figure 2(b).

**Perception** When watching a video, humans typically begin with a holistic observation of the scene and environment, and then shift attention to specific objects or events that appear abnormal. In line with this pattern, the perception stage of the proposed P2C-CoT reflects a transition from global observation to focused local observation. The model initially focuses on the whole environment, describes the scenes and recognizes the objects in the video. This step requires the model to have a comprehensive understanding of the normality in the video. Building upon this holistic understanding of the normality, the model then focuses on the events that deviate from the established normality, identifies what happens, when and where the event happens.

**Cognition** After observing the video content, humans typically identify abnormal events based on visual cues, and then proceed to reason about the potential consequences. Similarly, the cognitive stage of the proposed P2C-CoT reflects a progression from shallow cognition to deep cognition. The model first assesses the abnormality of the event and explains why it is considered anomalous with

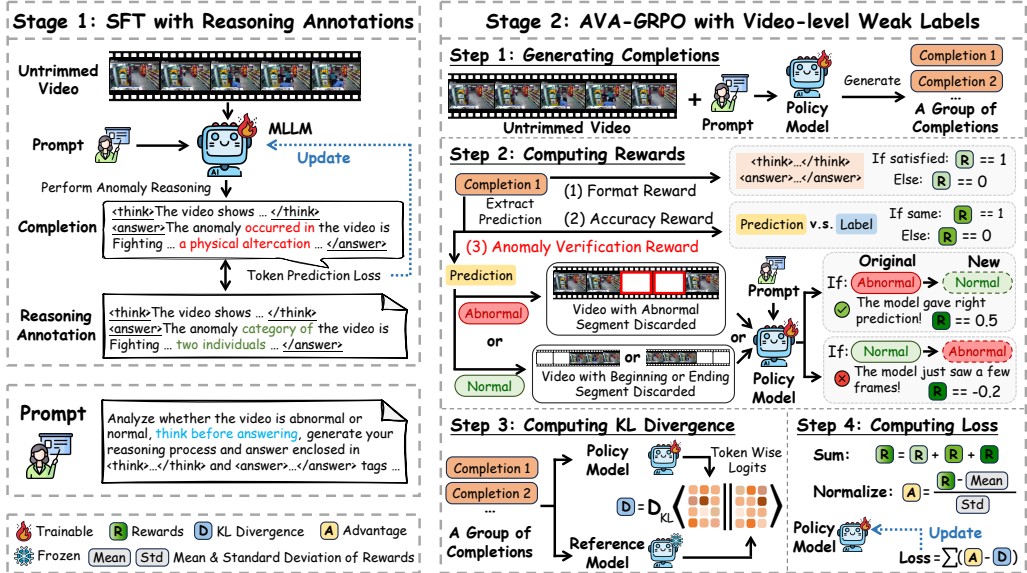

Figure 3: **Illustration of the two-stage training pipeline for Vad-R1.** Stage 1 enables the model to acquire basic reasoning capability with CoT annotated video. Stage 2 further enhances the model's reasoning capability through reinforcement learning.

relevant visual signals. It then engages in higher-level cognition to reason the underlying causes, the violated social expectations, and the possible consequences of the abnormal event.

**Answer** As shown in Figure 2(b), after completing the reasoning process, the model is expected to provide a concise summary of its judgment about the given video. The final answer consists of several key dimensions related to the anomaly, including category (**Which**), description of the event (**What**), spatio-temporal localization (**When** & **Where**), the reason **Why** it is identified as an anomaly and the potential influence (**How**). Notably, for normal videos, the corresponding P2C-CoT is simplified into two steps. Please refer to Appendix B for more details.

## 3.2 Dataset: Vad-Reasoning

**Video Collection** The existing VAD datasets generally lack the annotation of reasoning process. To construct a more suitable dataset for VAR, we take the following two aspects into consideration. On the one hand, we aim for the proposed dataset to cover a wide range of real-life scenarios. Similar to HAWK [54], we collect videos from current VAD datasets. The video scenarios include crimes under surveillance (UCF-Crime [53]), violent events under camera (XD-Violence [71]), traffic (TAD [39]), campus (ShanghaiTech [34]) and city (UBnormal [1]). Besides, we also collect videos from ECVA [12], a multi-scene benchmark. On the other hand, we strive to broaden the coverage of anomaly categories. To this end, we define a taxonomy of anomalies comprising three main types: Human Activity Anomaly, Environments Anomaly, and Objects Anomaly. Each type is categorized into several main categories, which are further divided into fine-grained subcategories. Then, we collect additional videos from the internet based on the existing dataset to expand the categories of anomalies. In total, the proposed Vad-Reasoning dataset contains 8203 videos for training and 438 videos for test. As shown in Figure 2(c), the training set of Vad-Reasoning is split into two subsets: Vad-Reasoning-SFT which contains 1755 videos annotated with high-quality reasoning process, and Vad-Reasoning-RL which contains 6448 videos with video-level weak labels.

**Annotation** To construct the proposed Vad-Reasoning dataset, we design a multi-stage annotation pipeline with two proprietary models Qwen-Max [59] and Qwen-VL-Max [61]. In order to ensure that the P2C-CoT annotation covers all key information in the video, we follow the principle of high frame information density [81]. Specifically, we first prompt Qwen-VL-Max to generate dense

description of video frames. These frame-level descriptions are then fed into Qwen-Max to generate the CoT step-by-step with different prompts. Please refer to Appendix B for more details.

## 3.3 AVA-GRPO

The original GRPO shows great effectiveness in text-based reasoning tasks. However, as mentioned above, the multimodal tasks like VAR are inherently more complex. In addition, there are only video-level weak labels available for RL stage due to high annotation costs, making it difficult to evaluate output quality based solely on accuracy and format reward. To address this challenge, we propose Anomaly Verification Augmented GRPO (AVA-GRPO), which introduces an additional reward through a self-verification mechanism, as illustrated in the right part of Figure 3.

**Overview of GRPO**   We begin by reviewing the original GRPO [50]. GRPO discards the value model and aims at maximizing the relative advantages among a group of generated answers. Given a question $q$, the model will first generate a group of candidate completions $O = \{o_i\}_{i=0}^{G}$. Subsequently, a set of rewards $R = \{r_i\}_{i=0}^{G}$ are computed based on the predefined reward functions. The rewards are then normalized within the group to compute the relative advantages as

$$A_i = \frac{r_i - \text{mean}(R)}{\text{std}(R)}, \tag{1}$$

where $A_i$ is the advantage score of $o_i$, which provides more effective assessment of both individual answer quality and relative comparisons within the group. What's more, to prevent the current policy $\pi_\theta$ from drifting excessively from the reference one $\pi_{\text{ref}}$, GRPO introduces a KL divergence regularization term. The final objective function of GRPO is formulated as

$$\mathcal{L}_{\text{GRPO}}(\theta) = \mathbb{E}_{\{q,O\}} \left[ \frac{1}{G} \sum_{i=1}^{G} \left( \min \left( \frac{\pi_\theta(o_i \mid q)}{\pi_{\theta_{\text{old}}}(o_i \mid q)} A_i, \text{clip} \left( \frac{\pi_\theta(o_i \mid q)}{\pi_{\theta_{\text{old}}}(o_i \mid q)}, 1 - \epsilon, 1 + \epsilon \right) A_i \right) \right. \right.$$
$$\left. \left. - \beta \, \mathbb{D}_{\text{KL}}(\pi_\theta \parallel \pi_{\text{ref}}) \right) \right], \tag{2}$$

where the ratio $\frac{\pi_\theta(o_i|q)}{\pi_{\theta_{\text{old}}}(o_i|q)}$ quantifies the relative change between the current policy and the old one, and the $\text{clip}(\cdot, 1 - \epsilon, 1 + \epsilon)$ operation constrains the ratio within a range.

**Anomaly Verification Reward**   GRPO replaces the value model with group relative scores, reducing the memory usage and training time. However, simple accuracy and format rewards are insufficient to evaluate the quality of answers for video anomaly reasoning task. To address this, we propose AVA-GRPO, an extension of GRPO that incorporates a novel anomaly verification reward. As shown in the right part of Figure 3, for each completion $o_i$, the predicted category of the video is first extracted. The video is then temporally trimmed based on the extracted prediction, and the trimmed video is fed into the model to generate a new answer. Additional anomaly verification rewards are assigned by comparing the original and regenerated answers.

On the one hand, if the video is initially classified as abnormal, the predicted temporal range of the abnormal event is extracted, and the corresponding segment is discarded from the original video to create a new trimmed video containing only normal segments. Then the trimmed video is re-fed into the model. If the trimmed video is subsequently predicted as normal, it suggests that the discarded segment is indeed abnormal and the model's initial prediction was correct. In this situation, a positive reward will be assigned to reinforce the model's original prediction.

On the other hand, inspired by Video-UTR [81], we consider the phenomenon of *temporal hacking* for video-MLLMs, where the models tend to generate predictions by relying only on a few frames, typically the beginning or ending of the video, instead of comprehensively processing the entire video sequence, which is detrimental to the recognition of anomaly events. As a consequence, if the video is initially predicted as normal, we randomly discard either the beginning or the ending segment of the video and feed the trimmed video into the model again. If the trimmed video is then

Table 1: Effectiveness of anomaly reasoning.

| Method | Strategy | Answer | | Detection | |
|---|---|---|---|---|---|
| | | BLEU-2 | METEOR | Recall | F1 |
| Qwen2.5-VL-7B [61] | Direct Answer | 0.184 | 0.339 | 0.431 | 0.597 |
| | Random Reasoning | 0.179 | 0.328 | 0.377 | 0.540 |
| | Structured Reasoning | **0.198** (+0.019) | **0.352** (+0.013) | **0.696** (+0.265) | **0.730** (+0.133) |
| Qwen3-8B [62] | Direct Answer | 0.038 | 0.184 | 0.368 | 0.534 |
| | Random Reasoning | 0.040 | 0.191 | 0.554 | 0.655 |
| | Structured Reasoning | **0.043** (+0.005) | **0.193** (+0.009) | **0.681** (+0.313) | **0.686** (+0.153) |
| Vad-R1 | Direct Answer | 0.268 | 0.441 | 0.838 | 0.861 |
| | Structured Reasoning | **0.293** (+0.025) | **0.487** (+0.046) | **0.843** (+0.005) | **0.862** (+0.001) |

predicted as abnormal, it suggests the model made its original prediction only based on insufficient visual evidence, which is not expected. Therefore, a negative reward is assigned in this case.

## 3.4 Training Pipeline

To address the challenge of lacking large-scale annotation, we design a two-phase training pipeline as shown in Figure 3. For the first stage, supervised fine-tuning is performed on the Vad-Reasoning-SFT dataset, in which videos are annotated with high-quality Chain-of-Thought (CoT) as described before. In this stage, the model's capability is gradually shifted from general multimodal understanding to video anomaly understanding, and it is enabled to acquire basic anomaly reasoning capability. In the second stage, training is continued on the Vad-Reasoning-RL dataset with the proposed AVA-GRPO reinforcement learning algorithm, which evaluates the quality of model responses in a self-verification manner with only video-level weak labels available. This stage aims at moving the model beyond pattern-matching tendencies from SFT, enabling it to develop more flexible, transferable anomaly reasoning capability. Please refer to Appendix C for more details.

## 4 Experiments

**Implementation Details**   Vad-R1 is trained with two stages based on Qwen-2.5-VL-7B [61]. For the first stage, SFT is performed with Vad-Reasoning-SFT dataset for four epochs. For the second stage, RL is performed with AVA-GRPO for one epoch, where there are only video-level weak labels available for Vad-Reasoning-RL dataset. All experiments are conducted with 4 NVIDIA A100 (80GB) GPUs. Please refer to Appendix C for more details.

**Evaluation Metrics and Baselines**   We first evaluate Vad-R1 on the test set of Vad-Reasoning, focusing on two aspects: anomaly reasoning and anomaly detection. For anomaly reasoning, we assess the text quality of reasoning process with BLEU [46], METEOR [3] and ROUGE [33] metrics. Besides, we also adopt LLM-as-judge evaluations [54]. For anomaly detection, we report accuracy, precision, recall and f1 scores for anomaly classification, along with mIoU and R@K for anomaly temporal grounding. Besides, to further explore the capabilities of Vad-R1, we also conduct experiments on VANE [15], a video anomaly benchmark for MLLMs, where the MLLMs are asked to answer multi-choice

Table 2: LLM-as-judge evaluations.

| Method | R | D | C |
|---|---|---|---|
| *Open-Source video MLLMs* | | | |
| InternVideo2.5 [69] | 0.580 | 0.517 | 0.487 |
| InternVL3 [96] | 0.692 | 0.608 | 0.586 |
| VideoChat-Flash [29] | 0.367 | 0.292 | 0.356 |
| VideoLLaMA3 [85] | 0.549 | 0.449 | 0.497 |
| LLaVA-NeXT-Video [94] | 0.541 | 0.452 | 0.491 |
| Qwen2.5-VL [61] | 0.638 | 0.555 | 0.542 |
| *Open-Source video reasoning MLLMs* | | | |
| Open-R1-Video [67] | 0.411 | 0.307 | 0.338 |
| Video-R1 [14] | 0.390 | 0.414 | 0.243 |
| VideoChat-R1 [30] | 0.634 | 0.559 | 0.528 |
| *MLLM-based VAD methods* | | | |
| Holmes-VAD [88] | 0.388 | 0.275 | 0.343 |
| Holmes-VAU [89] | 0.385 | 0.301 | 0.375 |
| HAWK [54] | 0.218 | 0.185 | 0.115 |
| *Proprietary MLLMs* | | | |
| Claude3.5-Haiku [2] | 0.711 | 0.637 | 0.611 |
| QVQ-Max [60] | 0.690 | 0.639 | 0.521 |
| GPT-4o [43] | 0.724 | **0.679** | 0.542 |
| **Vad-R1 (Ours)** | **0.734** | 0.659 | **0.662** |

questions. In this case, we report the accuracy of every category. We compare Vad-R1 with general video MLLMs [27, 32, 41, 87, 91], reasoning video MLLMs [30, 68, 14, 92] and some proprietary

Table 3: Performance comparison of anomaly reasoning and detection on Vad-Reasoning dataset.

| Method | Params. | Anomaly Reasoning | | | Anomaly Detection | | | | |
| --- | --- | --- | --- | --- | --- | --- | --- | --- | --- |
| | | BLEU-2 | METEOR | ROUGE-2 | Acc | F1 | mIoU | R@0.3 | R@0.5 |
| *Open-Source video MLLMs* | | | | | | | | | |
| InternVideo2.5 [69] | 8B | 0.110 | 0.264 | 0.109 | 0.715 | 0.730 | 0.417 | 0.458 | 0.424 |
| InternVL3 [96] | 8B | 0.124 | 0.286 | 0.116 | 0.779 | 0.756 | 0.550 | 0.613 | 0.540 |
| VideoChat-Flash [29] | 7B | 0.012 | 0.084 | 0.047 | 0.683 | 0.487 | 0.536 | 0.538 | 0.358 |
| VideoLLaMA3 [85] | 7B | 0.066 | 0.200 | 0.092 | 0.665 | 0.624 | 0.425 | 0.451 | 0.419 |
| LLaVA-NeXT-Video [94] | 7B | 0.094 | 0.238 | 0.104 | 0.651 | 0.423 | 0.576 | 0.601 | 0.585 |
| Qwen2.5-VL [61] | 7B | 0.113 | 0.264 | 0.116 | 0.761 | 0.730 | 0.567 | 0.610 | 0.563 |
| *Open-Source video reasoning MLLMs* | | | | | | | | | |
| Open-R1-Video [67] | 7B | 0.060 | 0.179 | 0.084 | 0.793 | 0.790 | 0.559 | 0.642 | 0.540 |
| Video-R1 [14] | 7B | 0.135 | 0.317 | 0.132 | 0.624 | 0.694 | 0.334 | 0.392 | 0.328 |
| VideoChat-R1 [30] | 7B | 0.128 | 0.287 | 0.123 | 0.793 | 0.790 | 0.559 | 0.642 | 0.540 |
| *MLLM-based VAD methods* | | | | | | | | | |
| Holmes-VAD [88] | 7B | 0.003 | 0.074 | 0.027 | 0.565 | 0.120 | - | - | - |
| Holmes-VAU [89] | 2B | 0.077 | 0.182 | 0.075 | 0.490 | 0.371 | - | - | - |
| HAWK [54] | 7B | 0.042 | 0.156 | 0.042 | 0.513 | 0.648 | - | - | - |
| *Proprietary MLLMs* | | | | | | | | | |
| Claude3.5-Haiku [2] | - | 0.097 | 0.253 | 0.098 | 0.580 | 0.354 | 0.518 | 0.543 | 0.524 |
| GPT-4o [43] | - | 0.154 | 0.341 | 0.133 | 0.711 | 0.760 | 0.472 | 0.565 | 0.476 |
| Gemini2.5-Flash [55] | - | 0.133 | 0.308 | 0.120 | 0.624 | 0.707 | 0.370 | 0.437 | 0.358 |
| *Proprietary reasoning MLLMs* | | | | | | | | | |
| Gemini2.5-pro [56] | - | 0.145 | 0.356 | 0.137 | 0.829 | 0.836 | 0.636 | 0.722 | 0.638 |
| QVQ-Max [60] | - | 0.142 | 0.318 | 0.121 | 0.702 | 0.747 | 0.430 | 0.503 | 0.412 |
| o4-mini [45] | - | 0.106 | 0.263 | 0.109 | **0.884** | **0.875** | 0.644 | 0.736 | 0.631 |
| **Vad-R1 (Ours)** | 7B | **0.233** | **0.406** | **0.194** | 0.875 | 0.862 | **0.713** | **0.770** | **0.706** |

models [60, 43, 56, 55]. Furthermore, we also consider MLLM-based VAD methods [54, 89, 88]. Due to space limitations, please refer to the Appendix D for more experimental results.

## 4.1 Main Results

**Does reasoning improve anomaly detection?** Table 1 demonstrates the effectiveness of anomaly reasoning. On the one hand, we evaluate the performance of Qwen2.5-VL [61] and Qwen3 [62]. As shown in the first two rows of Table 1, compared with directly answering, prompting models to reason according to the proposed perception-to-cognition chain-of-thought will gain greater performance. In the meanwhile, we evaluate the effect of random reasoning. In this case, the performance improvement is minimal, even inferior to direct answering. Notably, Qwen3 is a hybrid reasoning model that supports both reasoning and non-reasoning modes for the same task. The consistent performance gap across different settings further highlights the effectiveness of the proposed P2C-CoT for anomaly reasoning and detection. On the other hand, We compare the performance of Vad-R1 trained with the full P2C-CoT versus training with only the final answer portion of the P2C-CoT as shown in the third row of Table 1. When Vad-R1 is trained with only the final answer, it exhibits a performance drop.

**How well does Vad-R1 perform in anomaly reasoning and detection?** Table 2 and Table 3 shows the performance comparison on the test set of Vad-Reasoning. On the one hand, Table 2 shows the results of LLM-as-judge evaluations. Following HAWK [54], we evaluate the models' outputs based on Reasonability(R), Detail(D) and Consistency(C). We observe that Vad-R1 achieves the best performance among all open-source methods and even surpasses proprietary MLLMs such as GPT-4o, particularly in terms of Reasonability and Consistency. On the other hand, Table 3 shows the results of anomaly reasoning and detection tasks. Vad-R1 achieves great performance on both text quality of anomaly reasoning process and the accuracy of anomaly detection. It is worth noting that Vad-R1 significantly outperforms existing proprietary reasoning MLLMs Gemini2.5-Pro, QVQ-Max and o4-mini on anomaly reasoning capability, with BLEU score improvements of 0.088, 0.091, and

Table 4: Performance comparison on VANE.

| Method | SORA | OpenSORA | RG2 | VideoLCM | MS-T2 | Avenue | Ped1 | Ped2 |
|---|---|---|---|---|---|---|---|---|
| *Open-Source MLLMs* | | | | | | | | |
| Video-LLaMA [87] | 11.59 | 18.00 | 16.00 | 10.57 | 10.41 | 30.00 | 16.66 | 5.55 |
| VideoChat [27] | 10.74 | 28.00 | 4.00 | 17.64 | 20.83 | 32.25 | 13.33 | 13.88 |
| Video-ChatGPT [41] | 26.47 | 22.00 | 12.00 | 18.26 | 16.66 | 39.39 | 40.00 | 19.44 |
| Video-LLaVA [32] | 10.86 | 18.00 | 16.00 | 19.23 | 16.66 | 3.03 | 2.77 | 6.06 |
| MovieChat [51] | 8.69 | 10.00 | 16.00 | 14.42 | 6.25 | 18.18 | 6.66 | 11.11 |
| LLaMA-VID [31] | 7.97 | 14.00 | 20.00 | 19.23 | 14.58 | 27.27 | 6.66 | 19.44 |
| TimeChat [47] | 21.73 | 26.00 | 28.00 | 22.11 | 20.83 | 24.20 | 27.58 | 11.11 |
| *MLLM-based VAD methods* | | | | | | | | |
| Holmes-VAU [89] | 2.17 | 34.00 | 24.00 | 29.81 | 25.00 | 6.06 | 3.33 | 5.56 |
| Holmes-VAD [88] | 6.52 | 34.00 | 32.00 | 33.56 | 22.92 | 12.12 | 20.00 | 5.56 |
| HAWK [54] | 24.64 | 52.00 | 44.00 | 36.54 | 50.00 | 36.36 | 36.67 | 38.89 |
| **Vad-R1 (ours)** | **41.30** | **78.00** | **56.00** | **63.46** | **60.42** | **75.76** | **60.00** | **63.89** |

Table 5: Comparison of different training strategies for Vad-R1.

| Strategy | Anomaly Reasoning | | | | Anomaly Detection | | | | |
|---|---|---|---|---|---|---|---|---|---|
| | BLEU-2 | ROUGE-1 | ROUGE-2 | ROUGE-L | Prec. | mIoU | R@0.3 | R@0.5 | R@0.7 |
| Qwen2.5-VL | 0.113 | 0.505 | 0.199 | 0.477 | 0.768 | 0.567 | 0.610 | 0.563 | 0.526 |
| +SFT | 0.219 | 0.456 | 0.196 | 0.429 | 0.712 | 0.612 | 0.677 | 0.599 | 0.535 |
| +AVA-GRPO | 0.143 | 0.513 | 0.207 | 0.486 | 0.810 | 0.675 | 0.736 | 0.661 | 0.606 |
| +SFT+AVA-GRPO | **0.233** | **0.530** | **0.238** | **0.501** | **0.882** | **0.713** | **0.770** | **0.706** | **0.651** |

0.127, respectively. Besides, compared with existing MLLM-based VAD methods, Vad-R1 also exhibits greater advantages in anomaly reasoning and detection.

Table 4 shows the performance comparison on VANE benchmark. Vad-R1 achieves significant improvements and outperforms all baselines, including both general video MLLMs and MLLM-based VAD methods. Although the VANE benchmark is formulated as a multiple-choice question-answering task, its abnormal-related options inherently require the models to perform abnormal behavior recognition, semantic understanding, and reasoning. To better align with our anomaly reasoning setting, we also require the models to think before answering. Therefore, the evaluation on VANE serves as a complementary validation of Vad-R1's reasoning capability, demonstrating its superior reasoning ability under complex abnormal event understanding tasks.

## 4.2 Ablation Studies

**How to obtain the capability of reasoning?** Table 5 shows the effectiveness of different training strategies. When directly performing RL to the base model without prior SFT, the performance improvement is limited. This suggests that, without fundamental reasoning capability, the model struggles to benefit from RL training with video-level weak labels. In contrast, applying SFT leads to a more significant performance improvement, indicating that the structured Chain-of-Thought annotations effectively equip the model with basic anomaly reasoning capability. Notably, the combination of SFT and RL gains the best performance. The results align with the conclusion of DeepSeek-R1 [9], which suggests that SFT stage provides fundamental reasoning capability for the model, while RL stage further enhances its reasoning capability. Table 6 shows the effectiveness of the proposed AVA-GRPO. Compared with original GRPO, AVA-GRPO includes an additional anomaly verification reward, which incentivizes the anomaly reasoning capability of MLLM with only video-level weak labels. In addition, we add a length reward to control the length of output. For both 16-frame and 32-frame settings, AVA-GRPO outperforms the original GRPO across video reasoning and detection tasks. In contrast, using only one reward leads to limited or unstable improvement. These results demonstrate that the combination of length and anomaly rewards is essential for improving the overall reasoning and detection performance.

Table 6: Ablation results of different reward strategies.

| Frames | Strategy | Reasoning | | Detection | | | | |
|---|---|---|---|---|---|---|---|---|
| | | ROUGE-L | ROUGE-2 | Precision | mIoU | R@0.3 | R@0.5 | R@0.7 |
| 16 | GRPO | 0.502 | 0.475 | 0.861 | 0.712 | 0.770 | 0.699 | 0.640 |
| | GRPO+len_reward | 0.529 | 0.501 | 0.856 | 0.710 | 0.770 | 0.697 | 0.642 |
| | GRPO+ano_reward | 0.496 | 0.467 | 0.866 | 0.707 | 0.765 | 0.695 | 0.638 |
| | AVA-GRPO | **0.530** | **0.501** | **0.882** | **0.713** | **0.770** | **0.706** | **0.651** |
| 32 | GRPO | 0.495 | 0.468 | 0.831 | 0.695 | 0.761 | 0.692 | 0.624 |
| | GRPO+len_reward | 0.528 | 0.499 | 0.849 | 0.701 | 0.770 | 0.695 | 0.631 |
| | GRPO+ano_reward | 0.494 | 0.467 | 0.842 | 0.699 | 0.763 | 0.686 | 0.629 |
| | AVA-GRPO | **0.533** | **0.504** | **0.900** | **0.726** | **0.786** | **0.715** | **0.661** |

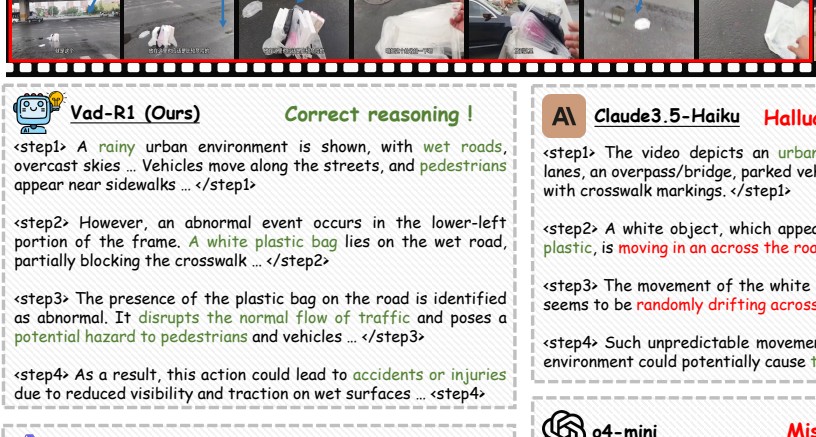

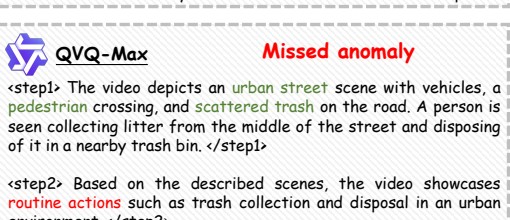

Figure 4: Qualitative result for an abnormal video.

## 4.3 Qualitative Analyses

As shown in Figure 4, Vad-R1 correctly performs anomaly reasoning and identifies the white plastic bag as an anomaly. In contrast, although Claude identifies the plastic bag as abnormal, it defines the cause of the abnormality as moving plastic bag, rather than the plastic bag acting as an obstacle. Besides, QVQ-Max and o4-mini also identify the white plastic bag, they do not treat it as an anomaly.

## 5 Conclusion

In this paper, we present Vad-R1, a novel end-to-end MLLM-based framework for video anomaly reasoning which aims to enable deep analysis and understanding of anomalies in videos. Vad-R1 performs structured anomaly reasoning process through a structured Chain-of-Thought that progresses gradually from perception to cognition. The anomaly reasoning capability of Vad-R1 is derived from a two-stage training strategy, combining supervised fine-tuning on CoT-annotated videos and reinforcement learning with an anomaly verification mechanism. Experimental results demonstrate that Vad-R1 achieves superior performance on anomaly detection and reasoning tasks.

# Acknowledgments

This work was supported in part by the National Key R&D Program of China (No.2022ZD0119200), the National Natural Science Foundation of China (No.62301621, No.62306343), Shenzhen Science and Technology Program (No. 20231121172359002, 2023A008), Shenzhen General Research Project (No. JCYJ20241202125904007), Guangdong Basic and Applied Basic Research Foundation (No. 2025A1515011398, No. 2025A1515011322), the CIE-Smartchip research fund (No.2024-08), and China Postdoctoral Science Foundation (No. 2025T180435, No. 2024M753741).

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

# A Summary of Appendix

This appendix provides supplementary information for the main paper. Firstly, we provide detailed information about the proposed Vad-Reasoning dataset, including the construction process, statistical analysis, and some examples. Then, we provide more experimental details covering prompts, settings, parameters, and computing resources. Furthermore, we provide more experimental results as well as visualizations. Finally, we discuss the potential impact and limitation.

# B The proposed Vad-Reasoning Dataset

## B.1 Annotation Pipeline

The training set of Vad-Reasoning consists of two subsets: Vad-Reasoning-SFT and Vad-Reasoning-RL. For Vad-Reasoning-RL, we retain the original dataset annotations and collapse them into video-level weak labels (Abnormal or Normal). For Vad-Reasoning-SFT, we design a multi-stage annotation process based on the proposed P2C-CoT, as shown in Figure 5.

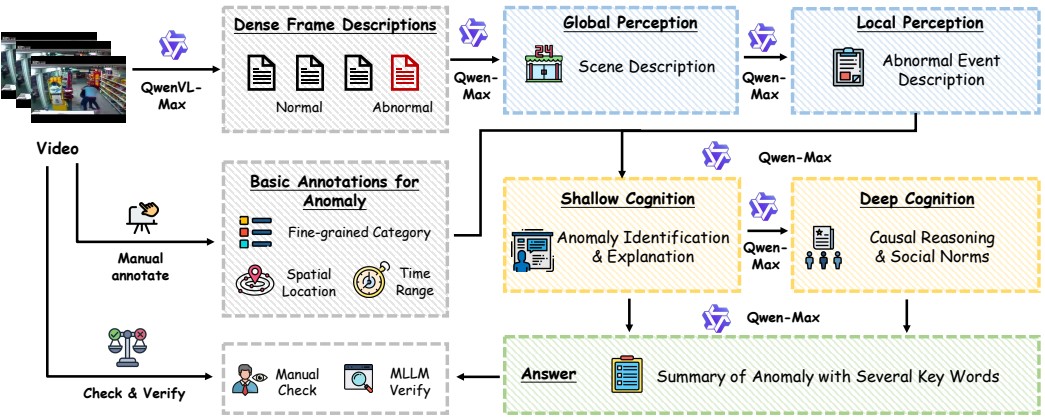

Figure 5: Illustration of multi-stage annotation process of Vad-Reasoning-SFT dataset.

Firstly, we manually annotate the fine-grained anomaly categories of the video, the duration of the anomaly events, and the approximate spatial location where the anomaly events occurred. Then, in order to ensure that the P2C-CoT annotation covers all key information in the video, we follow the principle of high frame information density [81]. Specifically, the video is decomposed into separate frames with a frame interval of 16. The extracted frames are then fed into Qwen-VL-Max to generate detailed dense frame descriptions. Considering the redundancy and high density of video information, directly prompting the model to generate annotations for the entire video would result in high annotation costs and information loss. In comparison, we first perform dense sampling at certain intervals, and then require Qwen-VL-Max to describe the video frames in detail, which can greatly preserve the key information in the video frames and reduce information loss.

After obtaining detailed dense video frame descriptions, we employ Qwen-Max to summarize and generalize the proposed P2C-CoT from these descriptions. Specifically, to ensure the structure of the reasoning process, we generate each step of P2C-CoT separately with different prompts, instead od generating them all at once. This design allows each step to focus on a specific task (e.g. step 1 focuses on scene description while step 2 focuses on abnormal event description). Each step will receive the output of previous step as input to ensure logical coherence. After generating all the reasoning steps, the model then generate a short summary about the anomaly in the video as final answer. In addition, we explicitly define anomaly categories to align model outputs with human-defined semantics, while enforcing constraints on relevance, objectivity, and neutrality. To further ensure the correctness of the generated CoT, all annotations are firstly verified by large language models and subsequently checked by human experts.

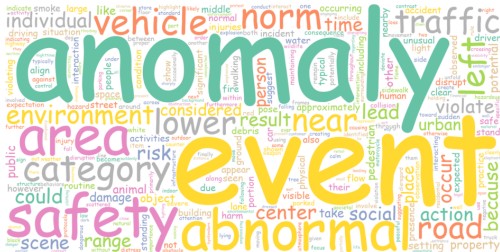
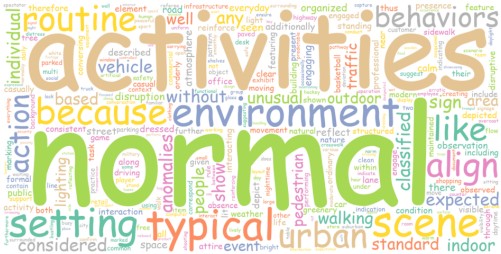

(a) Word cloud visualization for abnormal videos.    (b) Word cloud visualization for normal videos.

Figure 6: Word cloud visualizations of Vad-Reasoning-SFT dataset.

## B.2 Statistical Analysis and Comparison

We compare Vad-Reasoning with existing video anomaly detection and understanding datasets in Table 7 and Table 8. Vad-Reasoning consists of a total of 8641 videos, covering 34 million frames and over 360 hours of duration, making it one of the largest datasets among video anomaly understanding benchmarks. Besides, Vad-Reasoning-SFT provides fine-grained Chain-of-Thought (CoT) annotations, explicitly simulating human reasoning over abnormal events, with an average annotation length of 260 words. For the annotations, the recent video anomaly understanding datasets like CUVA [13] and ECVA [12] contain the description about the cause and effect of the anomaly. However, their corresponding annotations are isolated and disjointed, lacking a systematic structure and logical progression. In contrast, the proposed Vad-Reasoning-SFT datset provides structured and coherent anomaly reasoning annotation.

Figure 6 illustrates word cloud visualizations generated from the annotations of our Vad-Reasoning-SFT dataset. For abnormal videos, terms such as anomaly, event, and safety frequently appear, emphasizing that the CoT annotations of abnormal videos focus on irregular situations and potential risks. In contrast, normal videos are characterized by high-frequency words like activities, normal, and environment, which highlight everyday scenarios and typical human behaviors. Our Vad-Reasoning-SFT dataset not only captures descriptive content but also encourages models to engage in deeper causal interpretation of abnormal events. These observations indicate that our dataset is purposefully constructed to promote both recognition and reasoning in complex real-world contexts.

Figure 7 presents a comprehensive statistical overview of the proposed Vad-Reasoning dataset. The overall distribution of video length is relatively even as shown in Figure 7(a) and (b). Most of the videos in the Vad-Reasoning dataset are collected from UCF-Crime [53] and XD-Violence [71] as shown in Figure 7(c) and (d). And we collect additional 10 percent of videos from the internet. The proportion of normal and abnormal videos in the two subsets is basically balanced as shown in Figure 7(e). Finally, the fine-grained anomaly distributions are shown in Figure 7(f)-(h).

## B.3 Examples

We provide two examples of the proposed Vad-Reasoning dataset in Figure 8 and Figure 9. Notably, the CoT of normal videos will be simplified into two steps, the simple perception and cognition.

## C Implementation Details

### C.1 Prompt

The prompt used for performing video anomaly reasoning is shown in Figure 10. The prompt is composed of three parts, Task Definition , Output Specification and Format Requirements . Firstly, the Task Definition outlines the overall goal of video anomaly reasoning and explicitly require the model to think before answering. Secondly, the Output Specification provides detailed guidelines on the reasoning process and the expected answer. Finally, the Format Requirements presents concrete output examples with explicitly defined tags (e.g., <think></think> and <answer></answer>).

Table 7: Basic metadata comparison of datasets. Here "Mixture" indicates that the dataset is composed by integrating videos from multiple existing datasets.

| Dataset | Source | Videos | Frames | Duration | Resolution | FPS |
|---|---|---|---|---|---|---|
| **Traditional Video Anomaly Detection Datasets** | | | | | | |
| UCF-Crime [53] | Surveillance | 1900 | 13,741,393 | 128h | $320 \times 240$ | Multiple |
| XD-Violence [71] | Multiple | 4754 | 18,714,328 | 217h | Multiple | 24 |
| ShanghaiTech [34] | Campus | 437 | 317,398 | - | $856 \times 480$ | - |
| UCSD Ped1 [28] | Campus | 70 | 14,000 | - | $238 \times 158$ | - |
| UCSD Ped2 [28] | Campus | 28 | 4,560 | - | $360 \times 240$ | - |
| CUHK Avenue [37] | Campus | 37 | 30,652 | 0.3h | $640 \times 360$ | 25 |
| TAD [39] | Traffic | 518 | 540,212 | - | Multiple | - |
| UBnormal [1] | Generation | 543 | 236,902 | 2.2h | Multiple | 30 |
| NWPU Campus [5] | Campus | 547 | 1,466,073 | 16.3h | Multiple | 25 |
| **Video Anomaly Understanding Datasets** | | | | | | |
| UCA [82] | Surveillance | 1854 | 13,163,270 | 121.9h | $320 \times 240$ | Multiple |
| CUVA [13] | Multiple | 986 | 3,345,097 | 32.5h | Multiple | Multiple |
| ECVA [12] | Multiple | 2127 | 19,042,560 | 88.2h | Multiple | Multiple |
| VAD-Instruct50k [88] | Mixture | 6654 | 32,455,721 | 345h | Multiple | Multiple |
| HIVAU-70k [89] | Mixture | 6654 | 32,455,721 | 345h | Multiple | Multiple |
| HAWK [54] | Mixture | 7898 | 14,878,233 | 142.5h | Multiple | Multiple |
| **Vad-Reasoning-SFT** | Mixture | 2193 | 8,680,615 | 88.3h | Multiple | Multiple |
| **Vad-Reasoning-RL** | Mixture | 6448 | 25,495,729 | 272.2h | Multiple | Multiple |
| **Vad-Reasoning** | Mixture | **8641** | **34,173,344** | **360.5h** | Multiple | Multiple |

Table 8: The annotation type comparison of datasets. * denotes that the videos in Vad-Reasoning-RL are only labeled with video-level labels (Abnormal or Normal).

| Dataset | Anomalies | Text Annotation | Reasoning |
|---|---|---|---|
| **Traditional Video Anomaly Detection Datasets** | | | |
| UCF-Crime [53] | 13 | Anomaly class | - |
| XD-Violence [71] | 6 | Anomaly class | - |
| ShanghaiTech [34] | 13 | - | - |
| UCSD Ped1 [28] | 5 | - | - |
| UCSD Ped2 [28] | 5 | - | - |
| CUHK Avenue [37] | 5 | - | - |
| TAD [39] | 7 | - | - |
| UBnormal [1] | 22 | - | - |
| NWPU Campus [5] | 28 | - | - |
| **Video Anomaly Understanding Datasets** | | | |
| UCA [82] | 13 | Event descriptions | - |
| CUVA [13] | 42 | Anomaly description, cause, effect | Isolated |
| ECVA [12] | 100 | Anomaly description, cause, effect | Isolated |
| VAD-Instruct50k [88] | 13 | Clip caption & QA | - |
| HIVAU-70k [89] | 13 | Clip/Event/Video-level Caption & QA | Isolated |
| HAWK [54] | - | Anomaly description & QA | - |
| **Vad-Reasoning-SFT** | 37 | **Chain-of-Thought** | **Structured & coherent** |
| **Vad-Reasoning-RL** | 1* | Video-level label | - |
| **Vad-Reasoning** | 37 | Hybrid annotation | - |

## C.2 Training Process of AVA-GRPO

The core of the proposed AVA-GRPO is the additional anomaly verification reward as shown in Algorithm 1. Besides, we additionally consider a length reward. We first separately calculate the length of the reasoning text for abnormal videos and normal videos in Vad-Reasoning-SFT. During RL training, if the length of output satisfies the corresponding range, a length reward will be assigned. Notably, for each completion, the model will be only updated once. Consequently, the objective function of AVA-GRPO is simplified as

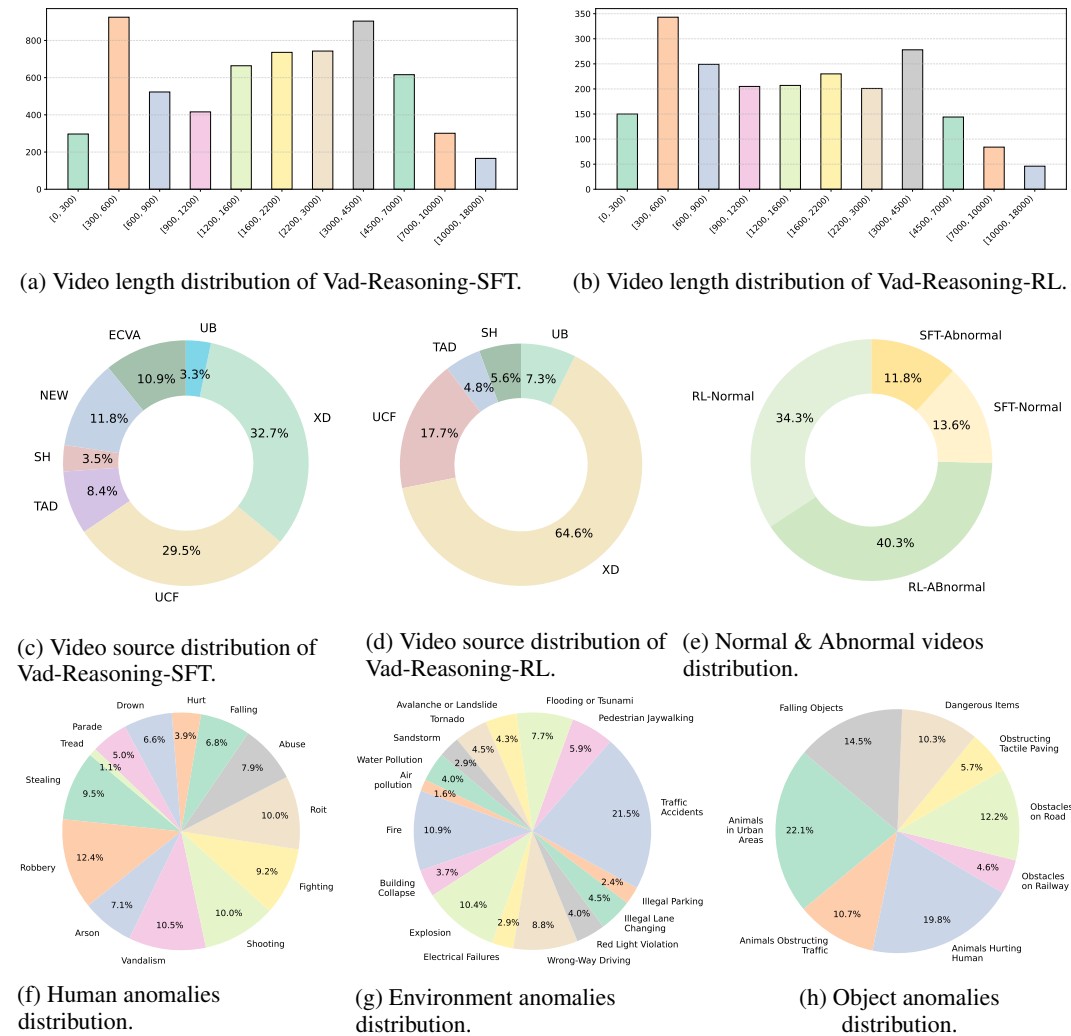

(a) Video length distribution of Vad-Reasoning-SFT.

(b) Video length distribution of Vad-Reasoning-RL.

(c) Video source distribution of Vad-Reasoning-SFT.

(d) Video source distribution of Vad-Reasoning-RL.

(e) Normal & Abnormal videos distribution.

(f) Human anomalies distribution.

(g) Environment anomalies distribution.

(h) Object anomalies distribution.

Figure 7: Statistical analyses of the proposed Vad-Reasoning dataset.

$$\mathcal{L}_{\text{AVA-GRPO}}(\theta) = \mathbb{E}_{\{q,O\}} \left[ \frac{1}{G} \sum_{i=1}^{G} \left( \frac{\pi_\theta(o_i \mid q)}{\pi_{\theta_{\text{no\_grad}}}(o_i \mid q)} A_i - \beta \, \mathbb{D}_{\text{KL}}(\pi_\theta \parallel \pi_{\text{ref}}) \right) \right], \tag{3}$$

where $\pi_{\theta_{\text{no\_grad}}}$ is equivalent to $\pi_\theta$. Finally, the training process of AVA-GRPO is shown in Algorithm 2.

## C.3 More Experimental Details

All experiments are conducted on 4 NVIDIA A100 (80GB) GPUs. For supervised fine-tuning stage, we train the base MLLM on Vad-Reasoning-SFT dataset for four epochs, taking approximately 6 hours. For reinforcement learning stage, we continue to train the model on the Vad-Reasoning-RL dataset for one epoch, taking about 26 hours. For efficiency, we uniformly normalize the video to 16 frames, and the maximum number of pixels per frame is limited to $128 \times 28 \times 28$ during training. The learning rates for both stages are set to $1 \times 10^{-6}$. The number of completions generated in a group is set to 4. The hyperparameter $\beta$ in Equation 3 is set as 0.04. AVA-GRPO includes five types of rewards. The specific values and meanings are shown in Table 9. For normal videos, the length range of reasoning process is set as $[140, 261]$, while it is set as $[233, 456]$ for abnormal videos.

**Step 1: Global Perception**

A young boy interacts with a large white cat in a calm indoor dining area. The cat sits on the boy's lap or nearby on a chair, and both face a window letting in natural light. The setting includes wooden furniture, plants, and framed pictures, creating a cozy atmosphere. The boy engages in routine behaviors like petting or sitting beside the cat.

**Step 2: Local Perception**

However, focusing on the abnormal event, we observe that in the lower-left portion of the frame, the boy leans forward toward the cat as if to kiss or nuzzle it. The cat remains seated on his lap during this interaction. This specific behavior introduces an element of unpredictability in their interaction.

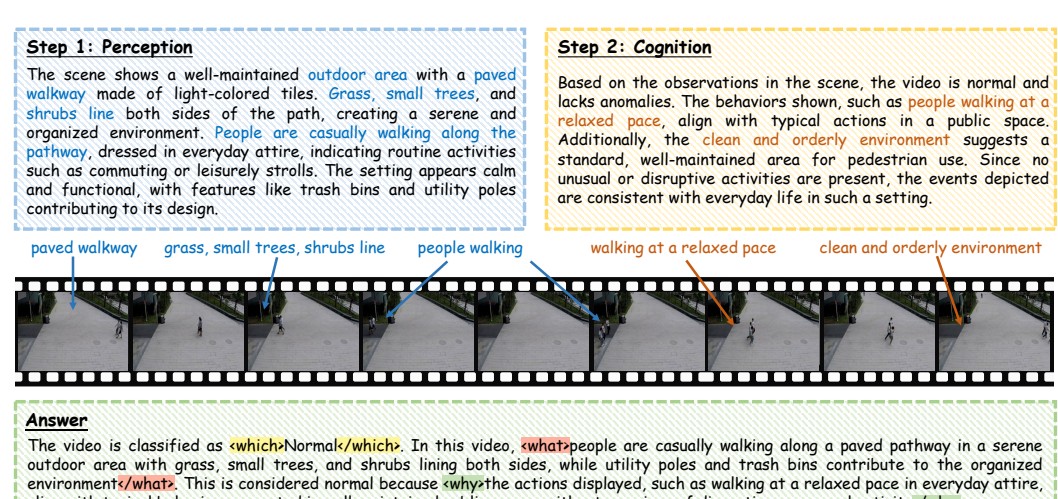

chair, wooden furniture    plants, pictures    kiss or nuzzle    lower-left portion of the frame

scratching or biting the boy    feel threatened    defensive reactions    suffer physical harm or emotional distress

**Step 3: Shallow Cognition**

Given this setup, the anomaly could involve the cat reacting suddenly, such as scratching or biting the boy when he leans closer. Such a reaction aligns with the \"Animals Hurt Human\" category, as the cat may feel threatened by the boy's forward-leaning action.

**Step 4: Deep Cognition**

This scenario violates social norms and safety practices, as approaching animals too closely without considering their body language can lead to defensive reactions. If the cat scratches or bites, the boy might suffer physical harm or emotional distress, emphasizing the importance of respecting animals' boundaries to prevent such incidents.

**Answer**

The anomaly category of the video is <which>Animals Hurt Human</which>. In this video, <what>a large white cat suddenly scratches or bites the boy who leans forward to nuzzle it</what>, occurring during the time range <when>[0.376, 0.564]</when>. This incident takes place in the <where>lower-left portion of the frame</where>, where the boy and the cat are interacting in an indoor dining area. This behavior is considered abnormal because <why>the cat may perceive the boy's action as a threat and react defensively, which deviates from the expected calm interaction</why>. Consequently, <how>such an event could result in physical injury or emotional distress for the boy, highlighting the violation of safety norms regarding animal handling and respect for their boundaries</how>.

Figure 8: An abnormal example of Vad-Reasoning.

**Step 1: Perception**

The scene shows a well-maintained outdoor area with a paved walkway made of light-colored tiles. Grass, small trees, and shrubs line both sides of the path, creating a serene and organized environment. People are casually walking along the pathway, dressed in everyday attire, indicating routine activities such as commuting or leisurely strolls. The setting appears calm and functional, with features like trash bins and utility poles contributing to its design.

**Step 2: Cognition**

Based on the observations in the scene, the video is normal and lacks anomalies. The behaviors shown, such as people walking at a relaxed pace, align with typical actions in a public space. Additionally, the clean and orderly environment suggests a standard, well-maintained area for pedestrian use. Since no unusual or disruptive activities are present, the events depicted are consistent with everyday life in such a setting.

paved walkway    grass, small trees, shrubs line    people walking    walking at a relaxed pace    clean and orderly environment

**Answer**

The video is classified as <which>Normal</which>. In this video, <what>people are casually walking along a paved pathway in a serene outdoor area with grass, small trees, and shrubs lining both sides, while utility poles and trash bins contribute to the organized environment</what>. This is considered normal because <why>the actions displayed, such as walking at a relaxed pace in everyday attire, align with typical behaviors expected in well-maintained public spaces without any signs of disruption or unusual activity</why>.

Figure 9: A normal example of Vad-Reasoning.

## C.4 Evaluation on VANE Benchmark

VANE [15] is a benchmark designed for evaluate the ability of video-MLLMs to detect anomalies in the video. It consists of 325 video clips and 559 question-answer pairs, covering both real-world surveillance and AI-generated video clips, and are categorized into nine anomaly types. For real-world anomalies, VANE collect 128 videos clips from existing video anomaly detection datasets (e.g., CUHK Avenue [37], UCSD-Ped1/Ped2 [28], and UCF-Crime [53]). For AI-generated anomalies, VANE includes 197 clips videos generated with SORA [4], OpenSora [16], Runway Gen2 [49], ModelScopeT2V [65] and VideoLCM [66]. We report the performance of Vad-R1 and other MLLM-based VAD methods on different categories. Notably, since Vad-R1 is trained with the proposed

Figure 10: Prompt template for performing video anomaly reasoning.

Table 9: Reward types and the corresponding values.

| Type | Meaning | Value |
|------|---------|-------|
| Accuracy | Evaluate classification result | 1 |
| Format | Evaluate format of output | 1 |
| Anomaly verification: Abnormal | Evaluate correctness of videos predicted as abnormal | 0.5 |
| Anomaly verification: Normal | Evaluate correctness of videos predicted as normal | -0.2 |
| Length | Evaluate length of output | 0.2 |

Vad-Reasoning dataset, which incorporates videos from UCF-Crime, we exclude the corresponding UCF-Crime subset from VANE benchmark.

# D   More Experimental Results

## D.1   LLM-Guided Evaluation

In this section, we provide additional LLM-guided evaluations [42, 95] to further assess the reasoning quality of Vad-R1. Table 10 demonstrates the results of pair-wise comparison [95]. Given the output of the our Vad-R1 and other models, we require GPT-4o [43] to compare the two answers and choose the better one. We also consider the impact of the order of answers on the judgment. The results show that GPT-4o overwhelmingly trends to choose Vad-R1 in both comparison orders, confirming the superior reasoning quality, contextual alignment, and interpretability of its responses. This consistent preference demonstrates that Vad-R1 not only produces semantically accurate descriptions but also exhibits a more human-like reasoning process. Table 11 shows the results of double-right [42], which jointly evaluates whether the model can provide a reasonable thinking process and correctly

---

**Algorithm 1** Anomaly verification reward

---

**Input**: Current video $v$, policy model $\pi_\theta$, generated completions $O = \{o_i\}_{i=1}^G$.
**Output**: Anomaly verification reward $R_{ano}$ .

1: Init anomaly verification reward: $R_{\text{ano}} = \{r_i\}_{i=1}^G$, where $r_i = 0$
2: **for** each $o_i \in O$ **do**
3:     Extract prediction $p$ of $v$ from completion $o_i$
4:     **if** $p ==$ Normal **then**
5:         Randomly discard either the beginning or the ending segment of $v$
6:     **else**
7:         Discard the predicted abnormal segment of $v$
8:     **end if**
9:     Obtain a trimmed video $\tilde{v}$
10:    Generate a new completion $\tilde{o} \sim \pi_\theta(\cdot \mid p, \tilde{v})$
11:    Extract new prediction $\tilde{p}$ of $\tilde{v}$ from new completion $\tilde{o}$
12:    **if** $p ==$ Abnormal and $\tilde{p} ==$ Normal **then**
13:       Assign positive reward $r_i \leftarrow 0.5$
14:    **else if** $p ==$ Normal and $\tilde{p} ==$ Abnormal **then**
15:       Assign negative reward $r_i \leftarrow -0.2$
16:    **end if**
17: **end for**
18: **return** $R_{\text{ano}} = \{r_i\}_{i=1}^G$

---

---

**Algorithm 2** AVA-GRPO

---

**Input**:Vad-Reasoning-RL dataset $\mathcal{D} = \{(v_j, Y_j)\}_{j=1}^N$, initial policy model $\pi_{\theta_{\text{init}}}$.
**Output**: Updated policy model $\pi_\theta$.

1: Init policy model: $\pi_\theta \leftarrow \pi_{\theta_{\text{init}}}$
2: Init reference model: $\pi_{\text{ref}} \leftarrow \pi_\theta$
3: **for** $e \in \{1, ..., E\}$ **do**
4:     **for** $(v_j, Y_j) \in \mathcal{D}$ **do**
5:         Generate a group of completions $O = \{o_i\}_{i=1}^G \sim \pi_\theta(\cdot \mid p, v_j)$
6:         Compute accuracy reward $R_{acc} = \{r_i\}_{i=1}^G$
7:         Compute format reward $R_f = \{r_i\}_{i=1}^G$
8:         Compute anomaly verification reward $R_{ano} \leftarrow$ **Algorithm 1**
9:         Compute length reward $R_{len} = \{r_i\}_{i=1}^G$
10:        Compute sum $R = R_{acc} + R_f + R_{ano} + R_{len}$
11:        Compute advantages $A = \frac{R - \text{mean}(R)}{\text{std}(R)}$
12:        Update $\pi_\theta$ with Equation 3
13:     **end for**
14: **end for**
15: **return** $\pi_\theta$

---

identify the anomaly, thereby offering a more comprehensive assessment of the anomaly reasoning capability. In this setting, we adopt semantic similarity to evaluate the correctness of the reasoning process. We adopt Qwen3-embedding [93] and Jina-embedding-V3 [52] to calculate semantic similarity separately. We observe that Vad-R1 achieves the highest RR score of 66.29% and 70.16% under different embedding models, indicating that its reasoning outputs are most aligned with the predictions.

## D.2 Experiments on More Input Tokens

During both training and inference, the video is uniformly sampled into 16 frames as input, with a maximum pixel count of $128 \times 28 \times 28$ per frame. In this section, we increase the number of frames to 32 and 64 per video, and the maximum pixel to $256 \times 28 \times 28$ per frame. The results are shown in Table 12. On the one hand, We observe that increasing the number of frame from 16 to 64 yields improvement across both anomaly reasoning and detection, showing that the extra frames

Table 10: Pair-wise comparison between Vad-R1 and other models.

| Model | | Rate | | | Model | | Rate | | |
|---|---|---|---|---|---|---|---|---|---|
| A | B | A_win | B_win | Tie | A | B | A_win | B_win | Tie |
| *Open-Source video MLLMs* | | | | | | | | | |
| **Vad-R1** | InternVideo2.5 [69] | **83.56** | 12.79 | 3.65 | InternVideo2.5 [69] | **Vad-R1** | 11.42 | **70.09** | 18.49 |
| **Vad-R1** | InternVL3 [96] | **78.82** | 17.31 | 3.87 | InternVL3 [96] | **Vad-R1** | 24.15 | **50.11** | 25.74 |
| **Vad-R1** | VideoChat-Flash [29] | **90.64** | 7.76 | 1.60 | VideoChat-Flash [29] | **Vad-R1** | 11.64 | **74.89** | 13.47 |
| **Vad-R1** | VideoLLaMA3 [85] | **85.93** | 10.09 | 3.98 | VideoLLaMA3 [85] | **Vad-R1** | 13.50 | **81.90** | 4.60 |
| **Vad-R1** | LLaVA-NeXT-Video [94] | **84.97** | 11.16 | 3.87 | LLaVA-NeXT-Video [94] | **Vad-R1** | 14.35 | **81.55** | 4.10 |
| **Vad-R1** | Qwen2.5-VL [61] | **83.83** | 11.16 | 5.01 | Qwen2.5-VL [61] | **Vad-R1** | 17.54 | **70.39** | 12.07 |
| *Open-Source video reasoning MLLMs* | | | | | | | | | |
| **Vad-R1** | Open-R1-Video [67] | **90.21** | 7.74 | 2.05 | Open-R1-Video [67] | **Vad-R1** | 5.24 | **93.62** | 1.14 |
| **Vad-R1** | Video-R1 [14] | **82.23** | 12.07 | 5.69 | Video-R1 [14] | **Vad-R1** | 8.88 | **86.10** | 5.01 |
| **Vad-R1** | VideoChat-R1 [30] | **79.95** | 14.81 | 5.24 | VideoChat-R1 [30] | **Vad-R1** | 18.68 | **70.62** | 10.71 |
| *MLLM-based VAD methods* | | | | | | | | | |
| **Vad-R1** | Holmes-VAD [88] | **80.28** | 5.57 | 14.15 | Holmes-VAD [88] | **Vad-R1** | 5.92 | **80.87** | 13.21 |
| **Vad-R1** | Holmes-VAU [89] | **78.54** | 8.68 | 12.79 | Holmes-VAU [89] | **Vad-R1** | 8.45 | **79.45** | 12.10 |
| **Vad-R1** | HAWK [54] | **77.45** | 4.33 | 18.22 | HAWK [54] | **Vad-R1** | 4.10 | **77.90** | 18.00 |

(a) Total reward.

(b) Std of total reward.

(c) Comparison on completion length.

(d) Comparison on error rate.

Figure 11: RL training curves of Vad-R1.

provide more useful visual evidence. On the other hand, the benefit of a higher resolution depends on the number of input frames. When increasing the max number of pixels to $256 \times 28 \times 28$ with 16 frames, the model gains small but consistent performance improvement, suggesting that high resolution details compensate for the short clip. In contrast, the performance will drop if we increase the max pixels for 32 frames, possibly due to token redundancy. Consequently, increasing frames is more useful, whereas higher resolution might lead to information overload.

### D.3 Training Curves

Figure 11 demonstrates the key training curves of Vad-R1 during RL stage. Figure 11(a) shows the total reward of AVA-GRPO, which increases steadily and converges after approximately 1000 steps, indicating consistent improvement in the degree of matching policy for the output of Vad-R1. Figure 11(b) illustrates the standard deviation of total reward, which decreases rapidly in the early stage and stabilizes below 0.1, suggesting that the output quality of Vad-R1 gradually improves as the training progresses. Figure 11(c) compares the completion length between GRPO and AVA-GRPO.

Table 11: Results of double-right metrics with different embedding models.

| Model | Qwen3-embedding | | | | Jina-embedding-V3 | | | |
|---|---|---|---|---|---|---|---|---|
| | RR↑ | RW↓ | WR↓ | WW↓ | RR↑ | RW↓ | WR↓ | WW↓ |
| *Open-Source video MLLMs* | | | | | | | | |
| InternVideo2.5 [69] | 37.59 | 25.51 | 2.05 | 34.85 | 41.91 | 21.18 | 7.97 | 28.94 |
| InternVL3 [96] | 50.57 | 27.33 | 0.23 | 21.87 | 56.72 | 21.18 | 1.14 | 20.96 |
| VideoChat-Flash [29] | 4.56 | 63.78 | 0.00 | 31.66 | 30.07 | 38.27 | 0.00 | 31.66 |
| Video-LLaMA3 [85] | 49.54 | 21.10 | 0.00 | 29.36 | 55.35 | 15.29 | 0.00 | 29.36 |
| LLaVA-NeXT-Video [94] | 41.23 | 23.92 | 0.23 | 34.62 | 52.62 | 12.53 | 0.91 | 33.94 |
| Qwen2.5-VL [61] | 50.34 | 25.74 | 0.00 | 23.92 | 56.49 | 19.59 | 1.37 | 22.55 |
| *Open-Source video reasoning MLLMs* | | | | | | | | |
| Open-R1-Video [67] | 26.42 | 37.59 | 0.00 | 35.99 | 38.95 | 25.06 | 0.68 | 35.31 |
| Video-R1 [14] | 11.85 | 50.57 | 0.23 | 37.35 | 30.75 | 31.66 | 6.83 | 30.76 |
| VideoChat-R1 [30] | 48.52 | 30.75 | 0.00 | 20.73 | 53.99 | 25.28 | 1.82 | 18.91 |
| *MLLM-based VAD methods* | | | | | | | | |
| Holmes-VAD [88] | 10.93 | 45.56 | 0.91 | 42.60 | 23.05 | 33.94 | 1.14 | 42.37 |
| Holmes-VAU [89] | 7.76 | 27.63 | 1.14 | 63.47 | 20.78 | 14.61 | 2.97 | 61.64 |
| HAWK [54] | 5.47 | 25.97 | 0.46 | 68.10 | 6.15 | 25.28 | 1.82 | 66.75 |
| *Proprietary MLLMs* | | | | | | | | |
| Claude3.5-Haiku [2] | 44.65 | 36.90 | 0.00 | 18.45 | 51.94 | 29.61 | 1.37 | 17.08 |
| GPT-4o [43] | 48.97 | 22.10 | 0.00 | 28.93 | 56.49 | 14.58 | 1.14 | 27.79 |
| Gemini2.5-Flash [55] | 34.02 | 28.54 | 0.00 | 37.44 | 39.73 | 22.83 | 2.05 | 35.39 |
| *Proprietary reasoning MLLMs* | | | | | | | | |
| Gemini2.5-Pro [56] | 32.65 | 50.46 | 0.00 | 16.89 | 60.27 | 22.83 | 0.00 | 16.90 |
| QvQ-Max [60] | 42.11 | 28.38 | 0.00 | 29.51 | 50.11 | 20.37 | 0.69 | 28.83 |
| o4-mini [45] | 57.37 | 31.34 | 0.00 | 11.29 | 66.36 | 22.35 | 0.23 | 11.06 |
| **Vad-R1 (Ours)** | **66.29** | 21.18 | 0.00 | 12.53 | **70.16** | 17.31 | 1.14 | 11.39 |

Table 12: Performance comparison of different numbers of input frames and spatial resolutions.

| Frames | Max Pixels | Anomaly Reasoning | | | Anomaly Detection | | | | |
|---|---|---|---|---|---|---|---|---|---|
| | | BLEU-2 | METEOR | ROUGE-2 | Acc | F1 | mIoU | R@0.3 | R@0.5 |
| 16 | $128 \times 28 \times 28$ | 0.233 | 0.406 | 0.194 | 0.875 | 0.862 | 0.713 | 0.770 | 0.706 |
| | $256 \times 28 \times 28$ | 0.238 | 0.412 | 0.198 | 0.886 | 0.878 | 0.713 | 0.772 | 0.702 |
| 32 | $128 \times 28 \times 28$ | 0.242 | 0.416 | 0.201 | **0.900** | 0.891 | **0.726** | 0.786 | **0.715** |
| | $256 \times 28 \times 28$ | 0.238 | 0.413 | 0.198 | 0.888 | 0.883 | 0.708 | 0.772 | 0.695 |
| 64 | $128 \times 28 \times 28$ | **0.244** | **0.420** | **0.203** | 0.895 | **0.892** | 0.709 | **0.777** | 0.695 |

We observe that the reasoning process generated by AVA-GRPO during the training process is shorter in length, presenting more concise and focused responses. Figure 11(d) presents the error rate comparison, where AVA-GRPO shows a clear downward trend and stabilizes at a lower level than vanilla GRPO. This indicates that AVA-GRPO effectively reduces incorrect reasoning and leads to more accurate and reliable responses during training.

## D.4 More Qualitative Results

We provide more qualitative results in Figure 13 and Figure 12. Vad-R1 demonstrates stable anomaly reasoning and detection capabilities. On the one hand, Figure 12 shows a normal example in Vad-Reasoning dataset. The video shows a normal scene in campus. Vad-R1 accurately describes the content in the video and identifies its normality. In comparison, GPT-4o and QVQ-Max demonstrate hallucination, pointing out anomalies that do not exist. And although Gemini2.5-Flash considers reverse walking as a potential abnormal event, it does not take into account the specific scenario of pedestrian walkways. On the other hand, Figure 13 shows results on VANE benchmark. When facing unseen videos, Vad-R1 still demonstrates great reasoning capability in complex environments and

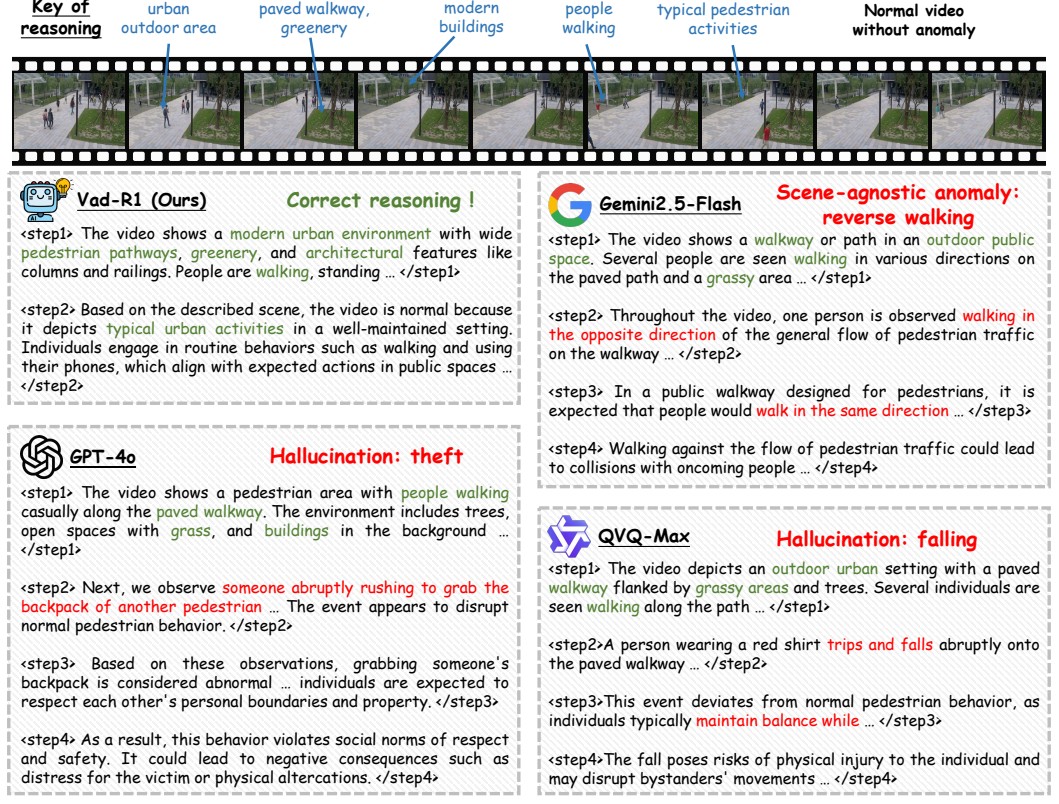

Figure 12: Qualitative result for a normal video.

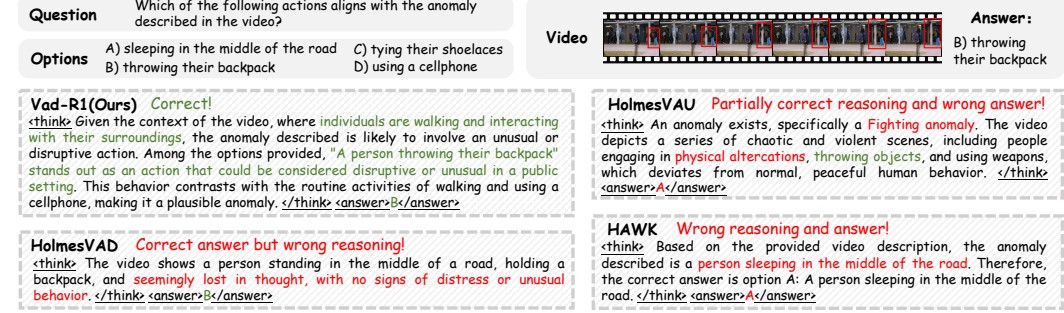

Figure 13: Qualitative performance on VANE benchmark.

correctly identifies anomalies in the video. In comparison, the reasoning process of HolmesVAU is partially correct, resulting in incorrect judgment, while HolmesVAD makes correct judgment but incorrect reasoning process.

# E   Impact and Limitation

In this paper, we propose a new task: Video Anomaly Reasoning, which enables MLLM to perform deep analysis and further understanding of the anomalies in the video. We hope our work can contribute to the video anomaly researches.

However, the inference speed of Vad-R1 remains a limitation, as the multi-step reasoning process introduces additional computational overhead.

