# OpenReview forum: "Vad-R1: Towards Video Anomaly Reasoning via Perception-to-Cognition Chain-of-Thought"
_NeurIPS.cc/2025/Conference — NeurIPS 2025 poster_

### Official Review · Reviewer_1Ake · 2025-07-02

**Clarity:** 2
**Significance:** 2
**Originality:** 2
**Rating:** 3
**Confidence:** 4

**Summary:**

The paper addresses the problem of video anomaly detection by leveraging the reasoning ability of multi-modal large language models. The idea is quite straightforward. The paper first curates a structured CoT dataset, then finetune a MLLM with this dataset using SFT, and finally train the MLLM using RL with verifiable reward. The results show the effectiveness of the proposed method on a few standard VAD benchmarks.

**Questions:**

See the weakness section.

**Ethical Concerns:**

["NO or VERY MINOR ethics concerns only"]

**Final Justification:**

The authors' rebuttal didn't fully convince me. I still think VAR and explainable VAD are fundamentally similar task.

- They are just of different interpretation format.

- I think the overall novelty is the application of RLVR to VAD. The contributions are rather limited, which is the major reason that I maintain my negative score.

**Quality:**

2

**Strengths And Weaknesses:**

Strengths

- The idea is simple and the presentation is clear.

- The training pipeline is pretty standard, consisting supervised finetuning and reinforcement learning with verifiable reward.

- The results look good.

Weakness

- The novelty is low. The paper essentially applies a standard reasoning elicitation approach (SFT, RLVR on long CoT data) to video anomaly detection. I fail to see any significant contribution beyong this simple application.

- A few baseline results are missing. For example, the author mentioned VERA (arXiv:2412.01095), but did not compare it in the experiment. There are quite a few MLLM-based VAD methods, and the paper didn't compare to them.

- The contribution of the long CoT data can be problematic. The annotation is fully dependent on the model output, without any refinement or debiasing strategies. Moreover, no ablation study of the dataset size and distribution is conducted.

---

> ### Author Rebuttal · Authors · 2025-07-31
>
> We sincerely appreciate your rigorous comments. While some critiques reflect fundamental concerns, we recognize their intent to enhance the paper's robustness. Below, we address each comment with analyses and clarifications.
> ***
> # Response to weakness 1: Clarification on contribution and novelty
> Our work is not just a simple application of the existing SFT+RL paradigm, it contains deep customization for video anomaly task, which is reflected in the following key aspects:
>
> * **Introduction of new task**: We propose a new task formulation, video anomaly reasoning (VAR), which moves beyond traditional video anomaly detection (VAD). Unlike VAD which focuses solely on whether an anomaly occurred, VAR requires models to generate causally complete natural language reasoning chains that explain what happened, why it happened, and which social norms were violated. This represents a substantial shift from perception to cognition in video understanding, and introduces new challenges for model design, supervision, and evaluation.
>
> * **Structured reasoning process**: Rather than prompting the model to simply "explain the video," we introduce a structured Chain-of-Thought in order to guide the model to perform structured anomaly reasoning. This explicit design enables the model to learn a stable and interpretable reasoning structure, and lays the foundation for both reward engineering and structured evaluation.
>
> * **Vad-Reasoning dataset**: The existing VAD datasets only provide video level labels or brief descriptions, lacking detailed reasoning annotations. The proposed Vad-Reasoning dataset is composed of two subsets. Vad-Reasoning-SFT (1755 videos with CoT annotations) for SFT cold start and Vad-Reasoning-RL (6448 videos with video level weak labels) for RL training. This design has scalability while saving annotation costs.
>
> * **Anomaly verification reward**: We propose AVA-GRPO, a novel variant of GRPO that introduces an anomaly verification reward, which requires the model to justify its predictions via video trimming and re-generation. It aims to enhance the model's anomaly reasoning ability without introducing additional annotations.
>
> In conclusion, our work is not a direct application of existing methods into the video anomaly domain. Instead, it presents substantive contributions centered around the novel video anomaly reasoning task, including innovations in task formulation, reasoning process design, dataset design and reinforcement learning algorithms.
> ***
> # Response to weakness 2: Clarification on missing baselines
>
> **The reasons for not choosing VERA**: VERA is essentially a traditional anomaly detection method while our Vad-R1 focuses on end-to-end video anomaly reasoning. VERA guides **frozen** MLLM to explain anomaly events in videos by learning a set of "trainable guiding questions". However, since the MLLM is frozen and no additional modules are introduced, if VERA is used as the baseline, it is equivalent to directly running its base model (InternVL2[r1] is used in its official code). In the main experiment, we have already compared the more advanced models of the same series (InternVideo2.5[r2] and InternVL3[r3]), so we did not take VERA as the baseline. Nevertheless, in response to the reviewer's suggestion, we add experiments comparing with VERA, and the results are as follows
>
> | Method | BLEU | METEOR | ROUGE | Acc | F1 | mIoU| R\@0.3 | R\@0.5 |
> |-|-|-|-|-|-|-|-|-|
> | VERA| 0.131 | 0.294 | 0.117  | 0.469 | 0.633 | 0.150 | 0.194  | 0.100 |
> | **Vad-R1** | **0.233** | **0.406**  | **0.194**  | **0.875** | **0.862** | **0.713** | **0.770**  | **0.706** |
>
> We observe that Vad-R1 achieves significantly better performance across all aspects, confirming that our end-to-end reasoning framework is more effective than detection-oriented pipelines like VERA.
>
> **Regarding other MLLM-based VAD methods**: On the one hand, although some methods employ MLLM, they still focus on traditional video anomaly detection. The MLLMs are only used as auxiliary modules. After the classifier gives the anomaly score, MLLM provides auxiliary explanations, such as AnomalyRuler[r4], VERA[r5] and LAVAD[r6]. In comparison, our Vad-R1 is an **end-to-end** anomaly reasoning model and does not require additional detection steps. On the other hand, some methods use MLLM for anomaly understanding. However, not all methods have open-sourced their code or model weights, such as Sherlock[r7] and AnomShield[r8]. So in the experiment, we only compare HAWK[r9], Holmes-VAD[r10] and Holmes-VAU[r11].
>
> In conclusion, within the scope of task type and reproducibility, we select the most representative and comparable MLLM-based method as baseline.
> ***
> # Response to weakness 3: Clarification on annotation quality
> **Data generation strategy**: We carefully design our annotation pipeline to mitigate noise and bias typically associated with LLM-generated data, which contains several key components:
>
> * **Step-wise generation**: The reasoning process is composed of four explicit steps. Rather than generating the entire reasoning chain in one shot, we generate each step independently with different dedicated prompts. This decomposition reduces error propagation and improves clarity and correctness.
>
> * **Explicit anomaly definition**: During generation, we explicitly provide pre-defined anomaly types to the proprietary model, ensuring that outputs align with our task design.
>
> * **Output constraints**: We add several output constraints into the prompts, including relevance, objectivity, and neutrality, to prevent hallucinations and ensure high-quality reasoning chains.
>
> * **Summary and calibration**: Given the information density in video content, each step may produce verbose outputs. To ensure clarity and conciseness, we introduce a final summary and calibration stage, where the proprietary model condenses each step into a refined 3–5 sentence paragraph. This guarantees that the final CoT remains focused and refined.
>
> * **Human verification**: After the data generation process, to ensure the quality of the data, we invite volunteers with backgrounds in video anomaly detection to manually check the generated annotations.
>
> **Data distribution and size ablation**: We perform ablation experiments on the data size of two training subsets.
>
> * **SFT**: The Vad-Reasoning-SFT dataset contains three major categories of anomaly: human anomaly, environment anomaly and object anomaly. We conduct ablation experiments on the categories of anomaly, and the results are as follows
>
> | Environment | Object | Human | BLEU  | ROUGE | METEOR | Recall |
> |-|-|-|-|-|-|-|
> | √|  | | 0.446  | 0.379  | 0.357  | 0.814  |
> |  | √ | | 0.426  | 0.359  | 0.344  | 0.485  |
> | | | √ | 0.456  | 0.376  | 0.362  | 0.701  |
> | √| √ | | 0.503  | 0.414  | 0.396  | 0.809  |
> | √|  | √ | 0.501  | 0.416  | 0.399  | 0.853  |
> |  | √ | √| 0.507  | 0.414  | 0.400  | 0.828  |
> | √| √ | √ | **0.513** | **0.455** | **0.411** | **0.971** |
>
> It can be observed that model trained with all categories achieves the best overall performance. These results confirm that diverse anomaly categories contribute positively to reasoning quality, and highlight the importance of balanced and comprehensive training data.
>
> * **RL**: We conduct ablation experiments on different data scales of Vad-Reasoning-RL dataset (30%, 60% and 100%). The results are as follows
>
> |Ratio | BLEU  | ROUGE | Precision | R\@0.5 | R\@0.7 |
> |-|-|-|-|-|-|
> | 30% | 0.507 | 0.427 | 0.734 | 0.638 | 0.546 |
> | 60% | 0.510 | 0.465 | 0.773 | 0.645 | 0.585 |
> | **100%** | **0.510** | **0.468** | **0.882** | **0.706** | **0.651** |
>
> The results show consistent performance improvement as the amount of RL data increases, highlighting that our dataset design supports practical expansion with minimal annotation cost.
>
> ***
> We deeply appreciate your rigorous comments, which has driven significant refinements to our methodology and interpretations. We will revise the paper accordingly and we are looking forward to further discussion during next phase.
>
> > [r1] Internvl: Scaling up vision foundation models and aligning for generic visual-linguistic tasks, CVPR 2024
> > [r2] Internvideo2. 5: Empowering video mllms with long and rich context modeling, arXiv:2501
> > [r3] InternVL3: Exploring Advanced Training and Test-Time Recipes for Open-Source Multimodal Models, arXiv:2504
> > [r4] Follow the Rules: Reasoning for Video Anomaly Detection with Large Language Models, ECCV 2024
> > [r5] VERA: Explainable Video Anomaly Detection via Verbalized Learning of Vision-Language Models, arXiv:2412
> > [r6] Harnessing Large Language Models for Training-free Video Anomaly Detection, CVPR 2024
> > [r7] Sherlock: Towards Multi-scene Video Abnormal Event Extraction and Localization via a Global-local Spatial-sensitive LLM, ACM WWW 2025
> > [r8] Exploring What Why and How: A Multifaceted Benchmark for Causation Understanding of Video Anomaly, arXiv:2412
> > [r9] Hawk: Learning to understand open-world video anomalies, NeurIPS 2024
> > [r10] Holmes-vad: Towards unbiased and explainable video anomaly detection via multi-modal llm, arXiv:2406.12235
> > [r11] Holmes-vau: Towards long-term video anomaly understanding at any granularity, CVPR 2025

---

> ### Comment · Area_Chair_xfAc · 2025-08-05
>
> Dear Reviewer 1Ake,
>
> As part of the post-rebuttal evaluation process, we kindly remind you to first review the authors' rebuttal and the other peer reviews, and then share your updated perspectives on the manuscript based on these considerations.
>
> Sincerely,
> AC

---

> ### Author Response · Authors · 2025-08-07
>
> Thank you once again for your comments on our paper. We would like to provide the following clarifications regarding your concerns about the novelty of our work.
>
> ---
>
> **VAR is fundamentally different from explainable VAD**
>
> Existing explainable VAD methods (e.g. VERA[r1] and Holmes-VAD[r2]) essentially first perform anomaly detection with traditional anomaly detector, followed by employing an MLLM to generate post-hoc descriptions for detected anomalies. In this setting, the detection and explanation are two **separate steps**. The MLLM merely provides supplementary information and does not participate in or influence the detection process itself.
>
> In contrast, the goal of our Video Anomaly Reasoning (VAR) is to require the MLLM to first reason about the entire video, and then produce a final decision and explanation about the anomaly. This is an **end-to-end process**, where the model receives only the video input and generates both the structured reasoning process and the final answer in a single forward pass. This formulation shifts the role of MLLMs **from passive describers to active reasoners and decision-makers**.
>
> ---
>
> **Our structured reasoning process is not just a special case of CoT**
>
> The general CoT is often free-from. However, video anomalies are inherently defined by human social and contextual norms. To reflect this, we explicitly simulate the human cognitive process and design the Perception-to-Cognition Chain-of-Thought, a four-stage reasoning schema tailored specifically for the VAR task.
>
> This structure is not only used during data annotation and training, but also directly aligned with evaluation and reward computation. Unlike generic CoTs, P2C-CoT is task-specific, and plays a role in all stages of the our method.
>
> ---
>
> **AVA-GRPO is more than a standard outcome-based RLVR reward**
>
> AVA-GRPO introduces a novel anomaly verification reward, which requires the model to justify its prediction by generating consistent outputs before and after trimming videos based on its own reasoning.
>
> This mechanism is not a generic outcome-based reward, it is a domain-specific solution to **the challenge of limited annotation**. Unlike prior MLLM-based VAD methods (e.g., HAWK[r3] and Holmes-VAD[r2]), which rely on large-scale annotation for anomaly understanding, our method requires only a small amount of high-quality reasoning annotations for SFT. During the RL stage, the model is trained using large-scale videos with only video-level labels. The anomaly verification reward thus enables efficient learning of reasoning ability **under weakly supervised settings**. Besides, the data in the RL stage can be practically expanded with minimal annotation cost, as described in our rebuttal.
>
> ---
>
> We sincerely appreciate your time and feedback. We would be glad to engage in discussions during the remainder of the discussion phase and look forward to your response.
>
>
> > [r1] VERA: Explainable Video Anomaly Detection via Verbalized Learning of Vision-Language Models, arXiv:2412
> > [r2] Holmes-vad: Towards unbiased and explainable video anomaly detection via multi-modal llm, arXiv:2406.12235
> > [r3] Hawk: Learning to understand open-world video anomalies, NeurIPS 2024

---

### Official Review · Reviewer_KKNF · 2025-07-02

**Clarity:** 3
**Significance:** 2
**Originality:** 2
**Rating:** 4
**Confidence:** 5

**Summary:**

This paper focuses on a new form of task based on video anomaly detection called video anomaly reasoning. This paper includes a new pipeline named P2C-CoT and a new dataset named Vad-Reasoning. Based on that, the paper conducts a training based on reinforcement learning called AVA-GRPO. The model is tested on anomaly reasoning and anomaly detection setting in this Vad-Reasoning as a text generation task. It also tests on the VANE benchmark.

**Questions:**

1. Are the used metrics objective and able to reflect the truthfulness of the reasoning in video anomaly reasoning? Why?
2. Performing R1-style training is interesting and would be the main technical contribution in this paper. However, is the training easy to run on Vad-Reasoning? What's the biggest obstacle in running R1 style training in the proposed dataset with MLLMs? What are the tricks we need to use to overcome the training difficulty? What's the additional cost for training on Vad-Reasoning like?
3. How's the ratio of training/test videos in Vad-Reasoning decided? The test video has a relatively small amount which looks unreasonable.

**Ethical Concerns:**

["NO or VERY MINOR ethics concerns only"]

**Final Justification:**

The authors have provided new results and a better explanation. The final score is adjusted based on that.

**Limitations:**

In addition to the additional computational overhead, the training cost can be limited and is not detailed in the paper.

**Quality:**

3

**Strengths And Weaknesses:**

Strengths:
1. The paper is well-motivated by improving the traditional video anomaly detection into a video anomaly reasoning task. This will help better improve MLLM's reasoning ability.
2. The proposed pipeline based on R1-style training is reasonable. This has been proved successful in training a MLLM with good reasoning and now it moves to the application of video anomaly reasoning.
3. The proposed dataset Vad-Reasoning is a good contribution to this field.

Weaknesses:
1. The used metrics like BLEU-2, METEOR, and ROUGE-2 are reasonable for text generation task. However, it lacks an explanation of whether they are good metrics for reflecting the true reasoning ability of MLLMs. This metric can be biased or inaccurate and the paper didn't explain whether they are suitable. Or in other wods, common ones used in video anomaly detection task is AUC, now we move to video anomaly reasoning and use those text generation metrics, are they really suitable? This is a question not answered in the paper.
2. Despite the introduction of Vad-Reasoning, whether it will be made open is not discussed in the paper or supplementary material. The number of training/test videos in the datasets look unreasonable, i.e., the number of test video is small compared to training videos.
3. Besides the Vad-Reasoning dataset, the proposed method is more like a direct application of DeepSeek-R1 training. The paper does have some customization for the task, but the novelty is limited.

---

> ### Author Rebuttal · Authors · 2025-07-31
>
> We greatly appreciate your valuable feedback. We also appreciate your recognition of our framework, training strategy, and dataset. We will respond to your questions in the following sections.
> ***
> # Response to weakness 1 & question 1:  Clarification on the effectiveness of evaluation metrics
> **Difference between tasks**: Traditional video anomaly detection focuses on identifying whether an anomaly occurs, typically measured via frame-level metrics like AUC or AP. In contrast, our proposed video anomaly reasoning aims to further analyze and understand the anomalies in the video, explaining the cause and consequence of anomalous events in natural language. This shift from simple detection to structured reasoning naturally calls for text quality evaluation.
>
> **Justification for text-based metrics**: We adopt BLEU, METEOR, and ROUGE to measure the lexical similarity between the generated and reference reasoning process, which are common in open-ended generation tasks and have also been used in prior MLLM-based video anomaly understanding studies such as HAWK[r1] and Holmes-VAU[r2].
>
> ***
> # Response to weakness 2 and question 3: Training/test split design and release plan of dataset
>
> **Design of training/test ratio**: To simulate real-world application scenarios, our training set is divided into two parts: Vad-Reasoning-SFT with a small number of detailed annotated videos, and Vad-Reasoning-RL with a large number of videos with only video level labels. The test set follows the same annotation format as the Vad-Reasoning-SFT subset. If we consider only the SFT portion of the training set, the ratio of test to training set is approximately 1:4, which is a standard and reasonable setting. The large-scale Vad-Reasoning-RL subset requires only a small amount of annotation cost and can be cheaply scaled in practical applications. The detailed statistical analysis of the dataset is provided in supplementary materials. We also conduct ablation experiments on different data scale of Vad-Reasoning-RL (30%, 60% and 100%). The results are as follows
>
> |Ratio | BLEU| ROUGE| Precision | R\@0.5 | R\@0.7 |
> |-|-|-|-|-|-|
> | 30% | 0.507 | 0.427 | 0.734 | 0.638 | 0.546 |
> | 60% | 0.510 | 0.465 | 0.773 | 0.645 | 0.585 |
> | **100%** | **0.510** | **0.468** | **0.882** | **0.706** | **0.651** |
>
> We observe that increasing the amount of RL data consistently improves model performance, demonstrating the effectiveness of our data structure design in enabling practical scalability.
>
> **Release plan**: We confirm that the Vad-Reasoning dataset, including the full reasoning annotations, will be released publicly in compliance with NeurIPS 2025 policies.
>
> ***
> # Response to weakness 3: Novelty of our work
> In this work, although the training paradigm of Vad-R1 draws on the idea of Deepseek-R1, Deepseek-R1 is a plain text model. In response to the unique requirements of the multimodal video anomaly detection tasks, we have made the following key customizations and innovations:
>
> **A challenging new Task**: Unlike traditional video anomaly detection, the proposed video anomaly reasoning aims to generate a causal and complete natural language reasoning chain from the video. This is an important expansion of the video anomaly task, and it also poses new challenges to the model structure and training paradigm.
>
> **Structured Chain of Thought**: Abnormal events are usually defined by humans. Therefore, we simulate the process of human thinking and propose the Perception-to-Cognition Chain-of-Thought, dividing the video anomaly reasoning process into four explicit stages. This design can guide the model to understand the anomalies step by step.
>
> **Customized reward design**: The proposed AVA-GRPO does not simply follow the existing RL method, but specifically constructs an **Anomaly Verification Reward** mechanism for the annotation limitation of the video anomaly reasoning task. It aims to enhance the model's abnormal reasoning ability without introducing additional annotations.
>
> Therefore, we believe that although the design of Vad-R1 is based on the general training paradigm, it has made targeted customizations, constituting a systematic innovation for the video anomaly task.
>
> ***
> # Response to question 2 and limitations: Feasibility of R1-style training
> Our main technical contribution lies in adapting and validating the R1-style training paradigm in the context of video anomaly task, which to our knowledge has not been explored in prior works. In the following, we explain the key obstacles and our solutions for R1-style training.
>
> **Key obstacles in training Vad-R1**:
>
> * **SFT cold start with structured reasoning data**: Unlike generic reasoning tasks, video anomaly reasoning requires the model to perform multi-step, structured reasoning. However, existing video anomaly detection datasets lack such annotations. Besides, starting RL directly from a base model leads to unstable training and suboptimal reasoning quality.
>
> * **Reward design under video level labels**: Given the high annotation cost, large-scale fine-grained annotation is unavailable. Thus, designing stable and effective reward functions is essential for enabling RL training under video level weak supervision.
>
> **Our solutions**:
>
> * **Perception-to-Cognition Chain-of-Thought**: We propose a novel structured anomaly reasoning chain, which simulates the human reasoning flow from perception to cognition. This provides a structured and interpretable format for supervision and evaluation.
>
> * **Vad-Reasoning dataset**: Based on the proposed CoT, we construct Vad-Reasoning-SFT dataset to support SFT training, where the videos are annotated with detailed reasoning process. In addition, we also construct a larger dataset Vad-Reasoning-RL for RL training, where videos only have video level weak labels.
>
> * **AVA-GRPO**: We extend original GRPO with an anomaly verification reward, which evaluates model outputs through self-consistency, promoting the anomaly reasoning ability of the model without additional annotations. Besides, a length reward is added to constrain the output length.
>
> **Training cost**: We keep the training cost practical. All training are conducted with four A100 GPUs. It takes about 6 hours and 30 hours for SFT and RL training, respectively. Considering the complexity of the video anomaly reasoning task and high cost of annotation, we believe the training cost is justified and acceptable.
>
> ***
> We extend our deepest gratitude to your comments again. All suggested revisions will be incorporated into the paper. We are looking forward to further discussion during the next phase.
>
> > [r1] HAWK: Learning to Understand Open-World Video Anomalies, NeurIPS 2024
> > [r2] Holmes-VAU: Towards long-term video anomaly understanding at any granularity, CVPR 2025

---

> > ### Comment · Reviewer_KKNF · 2025-08-05
> >
> > Dear Authors, thank you for your response! The new results are detailed and well-explained. The rating has been adjusted according to that.

---

> ### Comment · Area_Chair_xfAc · 2025-08-05
>
> Dear Reviewer KKNF,
>
> As part of the post-rebuttal evaluation process, we kindly remind you to first review the authors' rebuttal and the other peer reviews, and then share your updated perspectives on the manuscript based on these considerations.
>
> Sincerely,
> AC

---

> ### Author Response · Authors · 2025-08-06
>
> Dear Reviewer,
>
> Thank you for your feedback and taking the time to review our updated results.
>
> We hope that our responses have addressed all of your concerns. If there are any remaining issues that would benefit from further clarification, we would be happy to continue the discussion during the remaining time.
>
> Best regards,
>
> Authors of submission 21644

---

### Official Review · Reviewer_PJyt · 2025-07-04

**Clarity:** 3
**Significance:** 3
**Originality:** 3
**Rating:** 4
**Confidence:** 4

**Summary:**

This paper introduces Vad-R1, an end-to-end MLLM-based framework for Video Anomaly Reasoning (VAR). The authors propose a new task that goes beyond traditional video anomaly detection by requiring models to perform structured reasoning about anomalous events. The framework consists of three main components: (1) a Perception-to-Cognition Chain-of-Thought (P2C-CoT) that guides step-by-step reasoning from global perception to deep cognition, (2) the Vad-Reasoning dataset with fine-grained anomaly categories and reasoning annotations, and (3) a two-stage training pipeline combining supervised fine-tuning with a novel reinforcement learning algorithm called Anomaly Verification Augmented GRPO (AVA-GRPO).

**Questions:**

-	Dataset Annotation Methodology: While the use of proprietary models for annotation generation is understandable given the scale requirements, the paper would benefit from more detailed analysis of annotation quality. Some human validation samples or inter-model consistency analysis would strengthen confidence in the dataset quality.
-	AVA-GRPO Analysis Depth: Although the ablation study shows the effectiveness of the complete pipeline, a more detailed analysis of how the anomaly verification mechanism specifically contributes to performance would be valuable.

**Ethical Concerns:**

["NO or VERY MINOR ethics concerns only"]

**Final Justification:**

After carefully reading the authors response and comments by other reviewers, I decided to keep my original rating with an increased level of confidence. The authors have provided a comprehensive rebuttal, addressing key concerns of my reviews. While I agree on reviewer #VuPk on the limitation regarding the anomaly sample annotations, I believe the paper is valuable for the research community on the topic.

**Limitations:**

Yes

**Quality:**

3

**Strengths And Weaknesses:**

***Strengths***

-	The paper is well written.
-	The authors conduct extensive comprehensive comparisons against various baselines including general video MLLMs, reasoning-enhanced models, and proprietary systems, demonstrating consistent improvements.

***Weaknesses***

-	Dataset Quality Concerns: The paper's reliance on proprietary models (Qwen-Max and Qwen-VL-Max) for generating reasoning annotations is problematic. The authors provide insufficient analysis of annotation quality, consistency, or validation. There is no human evaluation of the generated reasoning chains, no inter-annotator agreement metrics, and no analysis of potential biases inherited from the proprietary models. This raises serious questions about the reliability of both the training data and evaluation benchmarks.
-	Limited AVA-GRPO Analysis: While the paper introduces AVA-GRPO as a key contribution, the ablation study in Table 4 only compares the full pipeline against individual components. There is no direct comparison between AVA-GRPO and standard GRPO on the same dataset, making it impossible to assess the specific contribution of the anomaly verification mechanism. The paper lacks analysis of how the verification rewards affect training dynamics or convergence.

---

> ### Author Rebuttal · Authors · 2025-07-31
>
> We are truly grateful for your insightful comments and your recognition of our experimental results and writing. In response to your concerns, we provide the following clarifications and explanations.
> ***
> # Response to weakness 1 & question 1:  Clarification on annotation quality
>
> **Data generation strategy**: To mitigate noise, inconsistency, and bias in LLM-generated data, we carefully designed a multi-stage annotation pipeline with the following components:
>
> * **Step-by-step generation**: The reasoning process is decomposed into four structured steps. Instead of generating the entire Chain-of-Thought (CoT) in one pass, each step is generated independently using tailored prompts. This design helps reduce error propagation and enhances the clarity and logical consistency of each reasoning step.
> * **Explicit anomaly definition**: To guide the generation process and ensure alignment with the anomaly definition, we provide explicit, pre-defined anomaly categories as input to the proprietary model. This reduces ambiguity and ensures relevance to the intended anomaly pattern.
> * **Output constraints**: We enforce constraints on model outputs, including relevance, objectivity, and neutrality, to minimize hallucinations and ensure the generated reasoning is grounded in video content.
> * **Summary and compression**: Given the high information density of videos, intermediate steps may result in verbose outputs. To address this, we introduce a refinement stage, where the model summarizes each reasoning step into concise 3–5 sentence paragraphs, improving readability and preserving essential content.
> * **Human verification**: To validate the quality of generated annotations, we conduct manual verification by volunteers with expertise in video anomaly detection. The volunteers check the outputs based on accuracy, coherence, and completeness.
>
> **Consistency analysis**: We conduct an inter-model consistency study. Specifically, we randomly select a subset of videos and apply the same prompt to generate reasoning process with different proprietary models. We then measure the semantic similarity between the generated annotations. The results are as follows
>
> | Similarity | Qwen-Max| GPT-4o | Gemini2.5 | Claude3.5 |
> |-|-|-|-|-|
> | **Qwen-Max** | 1 | 0.9059 | 0.8963 | 0.8948 |
> | **GPT-4o** | 0.9059 | 1 | 0.9068 | 0.8948 |
> | **Gemini2.5**| 0.8963 | 0.9068 | 1 | 0.8948 |
> | **Claude3.5**| 0.8948 | 0.8948 | 0.8948 | 1 |
>
> The results show that the outputs from different models exhibit high semantic alignment, suggesting that the generated annotations are not overly dependent on a specific model.
> ***
> # Response to weakness 2 & question 2: Further analyses of AVA-GRPO
>
> **Ablation study on rewards**: The proposed AVA-GRPO contains additional reward for anomaly verification and the reward to constrain output length. We conducted ablation experiments on these two rewards. Due to the length limit of the main text, we present the results in the Table 6 of supplementary materials. Some of the results are as follows
>
> |Strategy| ROUGE-L | ROUGE-2 | Precision | mIoU  | R\@0.3 | R\@0.5 | R\@0.7 |
> |-|-|-|-|-|-|-|-|
> | GRPO  | 0.502   | 0.475   | 0.861     | 0.712 | 0.770 | 0.699 | 0.640 |
> | GRPO+len_reward | 0.529   | 0.501   | 0.856| 0.710 | 0.770 | 0.697 | 0.642 |
> | GRPO+ano_reward | 0.496   | 0.467| 0.866 | 0.707 | 0.765 | 0.695 | 0.638 |
> | **AVA-GRPO**| **0.530** | **0.501** | **0.882** | **0.713** | **0.770** | **0.706** | **0.651** |
>
> It can be seen from the results that after introducing additional rewards, the performance of AVA-GRPO in anomaly reasoning and detection has been enhanced. Additionally, we can also observe that the performance improvement of adding only a single reward is unstable, which proves the effectiveness of the combination of two rewards.
>
> **Training dynamics**: We monitored and visualized the evolution trend of rewards and output lenght during the training process as shown in Figure 6 of supplementary materials. Besides, we compare the differences in training curves between the original GRPO and our proposed AVA-GRPO, and the specific results can be summarized as follows
>
> * On the **accuracy_reward** curve, we observe that the accuracy reward obtained by the AVA-GRPO model is significantly higher than that of the original GRPO.
> * On the **output_length** curve, we observe that the reasoning process generated by AVA-GRPO during the training process is shorter in length, presenting more concise and focused responses.
>
> In conclusion, AVA-GRPO not only optimizes the final performance but also demonstrates superior stability and efficiency during training.
>
> ***
>
> We sincerely appreciate your valuable feedback once again. We will revise the paper accordingly and we are looking forward to further discussion during the next phase.

---

> ### Comment · Area_Chair_xfAc · 2025-08-05
>
> Dear Reviewer PJyt,
>
> As part of the post-rebuttal evaluation process, we kindly remind you to first review the authors' rebuttal and the other peer reviews, and then share your updated perspectives on the manuscript based on these considerations.
>
> Sincerely,
> AC

---

### Official Review · Reviewer_VuPk · 2025-07-05

**Clarity:** 4
**Significance:** 3
**Originality:** 2
**Rating:** 4
**Confidence:** 5

**Summary:**

This paper proposes an MLLM-based method for Video Anomaly Reasoning (VAR) called Vad-R1 and a VAR dataset called Vad-Reasoning. Vad-Reasoning consists of videos from existing datasets and is annotated by the proposed Perception-to-Cognition Chain-of-Thought (P2C-CoT). Vad-R1 is trained with two stages: Supervised Fine-Tuning (SFT) with the P2C-CoT annotated videos and Reinforcement Fine-Tuning (RFT) with video-level weak labels. An improved RL algorithm called Anomaly Verification Augmented Group Relative Policy Optimization (AVA-GRPO) is proposed. Experiments show that Vad-R1 achieves superior VAR performance.

**Questions:**

1. What is the dataset for Table 1?
2. What is the loss function used for the SFT step?

**Ethical Concerns:**

["NO or VERY MINOR ethics concerns only"]

**Final Justification:**

The authors’ rebuttal has addressed some of my concerns; however, one of my primary concerns remains (Weakness 2): if well-annotated anomaly samples are used for training, Video Anomaly Reasoning becomes just a sub-problem of general video understanding reasoning. Therefore, I am raising my rating to 4: Borderline Accept (but not higher).

**Limitations:**

See weaknesses

**Quality:**

2

**Strengths And Weaknesses:**

Strengths
1. This paper applies the novel RFT technology to the VAR task and proposes a task-specific variant, which advances the research progress of the long-established VAD problem.
2. A new dataset, Vad-Reasoning is constructed, which is useful for training and benchmarking VAR models.
3. The experiments compare with multiple state-of-the-art video MLLMs, reasoning MLLM, and MLLM-based VAD approaches, which is comprehensive.
4. This paper is well-written, well-organized, and easy to follow. Its narrative reflects a deep understanding of the field and direction. Figures 2 & 3 help readers understand the proposed method.

Weaknesses:
1. There are several concerns regarding AVA-GRPO. The reward function considers only two cases: (1) a positive reward for an anomaly video (ground truth) where the prediction is anomaly (original) → normal (trimmed); and (2) a negative reward for an anomaly video (ground truth) where the prediction is normal (original) → anomaly (trimmed). In the first case, what if the prediction for the original video is correct, but the temporal localization is inaccurate? In this situation, the trimmed video may still contain abnormal frames. If the model incorrectly classifies the trimmed video as normal, it would still receive a positive reward. Furthermore, both cases assume the original video is abnormal, and there is no consideration for normal videos. This may lead to an imbalance in training signals.
2. It seems the training set includes well-annotated anomaly samples, another major concern. According to the standard formulation of anomaly detection, training should be either one-class (only normal data) or weakly supervised (normal data plus weakly labeled anomalies). If the model is trained on fully annotated anomaly samples, the task becomes closer to generic video understanding rather than true anomaly detection.
3. The metrics used for evaluating reasoning are insufficient. All of them rely on lexical comparisons. It is necessary to include semantic evaluation metrics, such as Doubly-Right [r1, 65] and LLM-as-a-Judge [r2].
4. According to Figure 4, the evaluation setting involves first performing detection, followed by answering a multiple-choice question (i.e., reasoning). This setup appears to match the one described in Line 34, which contradicts the stated motivation of this work.
5. According to Figure 4, the evaluation approach appears more aligned with action recognition than with VAD. This makes the claimed VAD capabilities less convincing.
6. (minor) The taxonomy of anomalies stated in L155 is similar to [65].
7. (minor) In L233, single choice questions -> multi-choice questions. In L98 and L108, Language Model -> Language Models.

Justification:

This paper applies the most advanced technology to the VAD task, which is a good contribution. However, there are concerns about the proposed method, problem formulation, and evaluation. I suggest Borderline Reject at this stage.

[r1] Mao, Chengzhi, et al. "Doubly right object recognition: A why prompt for visual rationales." CVPR, 2023.

[r2] Zheng, Lianmin, et al. "Judging llm-as-a-judge with mt-bench and chatbot arena." NeurIPS, 2023.

---

> ### Author Rebuttal · Authors · 2025-07-31
>
> We are deeply grateful for your thoughtful feedback and valuable suggestions. We appreciate your recognition of our framework, dataset, experiments, and writing. In the following, we carefully respond to each comment and provide additional explanations.
> ***
> # Response to weakness 1: Concerns of AVA-GRPO's implementation details
> **Reward bias caused by inaccurate temporal localization**: We acknowledge that inaccurate predictions of the temporal boundaries of anomalies may potentially affect the reward calculation. However, we emphasize that AVA-GRPO is essentially a self-supervised verification mechanism, which is designed to approximately evaluate the correctness of temporal localization under weak supervision. In practice, we first perform uniform sampling on the video due to the contextual limitation. When computing the reward, we perform the trimming operation on the raw untrimmed video, followed by resampling, rather than trimming on the already sampled frame set. This ensures that even with inaccurate localization, the trimmed video can still contain enough abnormal frames to support detection, preserving the robustness of the reward signal.
>
> **The absence of explicit rewards for normal videos**: The AVA-GRPO is biased toward reinforcing anomaly understanding, which aligns with the nature of task. The MLLM has acquired general video understanding capabilities during the **pre-training** stage. And the anomalies are often human-defined, thus we hope that the model can learn the pre-defined anomaly patterns during **post-training** (SFT+RL) stage. Additionally, our training set also includes normal videos, ensuring the model retains an understanding of normal patterns and avoids catastrophic forgetting.
> ***
> # Response to weakness 2: Clarification with additional annotations
> **Differences in Tasks**: We do not aim to solve traditional anomaly detection, but rather address a novel and more challenging task—Video Anomaly Reasoning (VAR), which aims to not only determine whether a video is abnormal, but also to gradually infer complex semantic information. In this task setting, the model needs to possess causal reasoning capabilities similar to those of humans, which cannot be learned merely through traditional VAD Settings (such as only normal data or video-level labels). Therefore, we simulate the process of human thinking and introduce the Perception-to-Cognition Chain-of-Thought. Based on this, we construct a structured annotated dataset (Vad-Reasoning-SFT) for the first stage of supervised fine-tuning (SFT) to help the model master the basic reasoning logic.
>
> **Our training strategy**: The training pipeline of Vad-R1 is divided into two stages: (1) The first stage (SFT) uses **a small number** of videos (1755) with detailed reasoning annotations, (2) The second stage (RL) involves a **large-scale** dataset (6448 videos) that contains only video-level weak labels. Therefore, the overall training strategy remains consistent with the real-world challenge conditions of anomaly detection, where fully annotated anomaly samples are scarce, and models must learn from weak supervision.
> ***
> # Response to weakness 3: Additional evaluation metrics
> We agree the limitations of using only lexical-level metrics (e.g., BLEU, METEOR, ROUGE) for evaluating reasoning quality. Therefore, referring to HAWK[r1], we evaluated the reasoning results with the proprietary model in three aspects: Reasonability, Detail, and Consistency. Due to the length limitation of the main text, the complete results are presented in Table 4 of the supplementary materials. Some of the results are shown below:
> |Method|Reasonability|Detail|Consistency|
> |-|-|-|-|
> | Holmes-VAD |0.388|0.275| 0.343 |
> | Holmes-VAU |0.385 | 0.301| 0.375 |
> | HAWK | 0.218 | 0.185 | 0.115 |
> | Claude3.5-Haiku  | 0.711 | 0.637 | 0.611 |
> | QvQ-Max | 0.690 | 0.639 | 0.521 |
> | GPT-4o | 0.724 | **0.679** | 0.542 |
> | **Vad-R1 (Ours)**| **0.734** | 0.659 | **0.662** |
>
> It can be observed that Vad-R1 demonstrates superior performance, particularly in terms of Reasonability and Consistency. In addition, based on the reference paper you provided, we supplement the following experiments:
>
> **Doubly-Right**: In this setting, we adopt similarity to evaluate the correctness of the reasoning process. We use qwen3-embedding[r2] to encode text to calculate similarity. The results are as follows
> | Model | RR ↑ | RW ↓ | WR ↓ | WW ↓ |
> |-|-:|-:|-:|-:|
> | Holmes-VAD|10.93 |45.56 |0.91 |42.60 |
> | Holmes-VAU |7.76 |27.63 |1.14 |63.47 |
> | Hawk | 5.47 | 25.97 |0.46 |68.10 |
> | Claude-3-5-haiku | 44.65 | 36.90 | 0.00 |18.45 |
> | GPT-4o |48.97 |22.10 |0.00 |28.93 |
> | Gemini-2.5-flash |34.02 |28.54 |    0.00 |   37.44 |
> | Gemini-2.5-pro |32.65 |50.46 |    0.00 |   16.89 |
> | QVQ-max|42.11 |   28.38 |0.00 |   29.51 |
> | o4-mini| 57.37 |   31.34 |0.00 |   11.29 |
> | **Vad-R1 (Ours)**| **66.29** | 21.18 | 0.00 | 12.53 |
>
> We observe that Vad-R1 achieves the highest RR score of 66.29%, indicating that its reasoning outputs are most aligned with the predictions.
>
> **LLM as Judge**: In this setting, we consider pairwise comparison. We choose GPT-4o as the judge, providing it with the outputs of Vad-R1 and other models, and requiring it to choose a better answer. We also consider the impact of the order of answers on the judgment. The results are as follows
>
> | Model_A  | Model_B  | A_win_rate | B_win_rate | Tie_rate |
> |-|-|-|-|-|
> | **Vad-R1**  | Hawk | **77.45**  | 4.33 | 18.22 |
> | **Vad-R1** | Holmes-VAD | **80.28**  | 5.57 | 14.15 |
> | **Vad-R1**  | Holmes-VAU | **78.54**  | 8.68| 12.79|
> | Hawk | **Vad-R1** | 4.10 | **77.90**  | 18.00 |
> | Holmes-VAD  | **Vad-R1** | 5.92 | **80.87**  | 13.21 |
> | Holmes-VAU  | **Vad-R1** | 8.45 | **79.45**  | 12.10 |
>
> We observe that Vad-R1 is significantly superior to other methods, and GPT-4o tends to choose the generated results of Vad-R1 under two conditions.
>
> ***
> # Response to weakness 4 & 5: Clarification of the evaluation setting
> **Evaluation setting**: Our evaluation consists of two aspects: on our test set (Table 1, 2, and 4) and the VANE benchmark (Table 3 and Figure 4). On the test set, the model performs standard anomaly reasoning (thinking before answering).  Figure 4 illustrates an example from the VANE benchmark, which is designed in the form of "video + multiple-choice question." In this setting, the model directly receives the entire video, the question and the options. The model first thinks (enclosed with \<think> tags) and then answers the given question (enclosed with \<answer> tags), which is essentially different from the one described in Line 34. Our model does not rely on any external modules but completes reasoning and decision-making **end-to-end**. Besides, due to the length limitation of main text, we provide the qualitative results of evaluation on our test set in supplementary materials.
>
> **Evalutaion method on VANE**: The purpose of the VANE benchmark is to evaluate the understanding ability of MLLMs for abnormal events. Although it is a "multi-choice question" in form, the abnormal content involved in its options requires the model to have the ability of abnormal behavior recognition, semantic understanding and reasoning. So we regard the evaluation on the VANE benchmark as a supplementary way to demonstrate the performance of Vad-R1 under complex reasoning tasks.
> ***
> # Response to weakness 6: Difference with AnomalyRuler on taxonomy of anomalies
>
> Both our Vad-R1 and AnomalyRuler[r3] indeed categorize anomalies from the perspectives of human activities and environmental/contextual elements. But there are several key distinctions:
>
> **Purpose of Taxonomy**: AnomalyRuler uses the taxonomy primarily for rule induction from few-shot normal frames.  In contrast, our taxonomy aims to cover possible anomalies in real life. Based on the taxonomy, we can build our dataset and further construct the reasoning annotation.
>
> **Granularity and application**: We define fine-grained subcategories across multiple sources, supporting high-quality reasoning annotations and enabling both detection and open-ended causal reasoning. AnomalyRuler's rules are relatively abstract.
> ***
> # Response to weakness 7: Inappropriate expression
> Thank you for pointing out the inappropriate expression. We will correct the issues in the revised version.
> ***
> # Response to question 1 & 2: Unclear descriptions
> **Dataset for Table 1**: The experiments in Table 1 are conducted on the test set of Vad-Reasoning we proposed (the same as those in Table 2 and 4).
>
> **Loss function for SFT**: The SFT stage employs the standard Token-level Cross-Entropy Loss, which maximizes the likelihood probability of the generated reasoning process. It can be formulated as:
> $$
> L=-\sum_{t=1}^T \log P(y_t \mid y_{<t}, x)
> $$
> where $y_t$ is the ground truth token at time step $t$, $y_{<t}$ is the sequence of previous tokens, and $x$ denotes the multimodal input.
> ***
> We sincerely appreciate your comments once again and will revise the paper accordingly. We are looking forward to further discussion during next phase.
>
> > [r1] HAWK: Learning to Understand Open-World Video Anomalies, NeurIPS 2024
> > [r2] Qwen3 Embedding: Advancing Text Embedding and Reranking Through Foundation Models, arXiv 2506
> > [r3] Follow the Rules: Reasoning for Video Anomaly Detection with Large Language Models, ECCV 2024

---

> ### Comment · Area_Chair_xfAc · 2025-08-05
>
> Dear Reviewer VuPk,
>
> As part of the post-rebuttal evaluation process, we kindly remind you to first review the authors' rebuttal and the other peer reviews, and then share your updated perspectives on the manuscript based on these considerations.
>
> Sincerely,
> AC

---

> ### Comment · Reviewer_VuPk · 2025-08-09
>
> Thank you for your responses.
>
> The authors’ rebuttal has addressed some of my concerns; however, one of my primary concerns remains (Weakness 2): if well-annotated anomaly samples are used for training, Video Anomaly Reasoning becomes just a sub-problem of general video understanding reasoning. Therefore, I am raising my rating to 4: Borderline Accept (but not higher).
>
> Please incorporate the responses into the revised version, which will make the paper stronger.

---

> ### Author Response · Authors · 2025-08-09
>
> Thank you for your follow-up and acknowledging part of our clarification. We understand your concern about additional annotations. We emphasize that Video Anomaly Reasoning (VAR) fundamentally differs from traditional anomaly detection or existing anomaly understanding. It requires the model to progressively reason from perception to cognition, identifying the event, its cause, consequences, and the reason it violates norms. Such causal and normative reasoning **cannot be learned from conventional annotations alone**.
>
> Besides, unlike prior MLLM-based VAD methods such as HAWK[r1] and Holmes-VAD[r2] that rely on large-scale annotated videos for anomaly understanding, we annotate only **a small subset** to bootstrap basic reasoning capability in stage 1 (SFT). The core training happens in stage 2 (RL) with **a much larger set** containing only video-level labels, where the proposed AVA-GRPO incentivizes reasoning capability **under weak supervision**.
>
> We sincerely thank you again for your feedback and would be happy to further discuss or clarify during the remaining discussion period.
>
> > [r1] Hawk: Learning to understand open-world video anomalies, NeurIPS 2024
> > [r2] Holmes-vad: Towards unbiased and explainable video anomaly detection via multi-modal llm, arXiv:2406.12235

---

> > ### Comment · Reviewer_VuPk · 2025-08-09
> >
> > Thanks for following up. I'm keeping my updated rating as 4.

---

### Decision · Program_Chairs · 2025-09-17

**Decision:**

Accept (poster)

**Comment:**

This paper has received 3 weak accept ratings and 1 weak reject rating. The authors’ rebuttal has addressed most of the issues raised by the reviewers. Video Anomaly Reasoning is a highly interesting innovation, and the proposed dataset also contributes positively to research in this direction. Therefore, I recommend accepting the paper.